# Modelling transmission and control of the COVID-19 pandemic in Australia

Sheryl L. Chang [1], Nathan Harding [1], Cameron Zachreson [1], Oliver M. Cliff [1] & Mikhail Prokopenko [1,2✉]

There is a continuing debate on relative benefits of various mitigation and suppression strategies aimed to control the spread of COVID-19. Here we report the results of agent-based modelling using a fine-grained computational simulation of the ongoing COVID-19 pandemic in Australia. This model is calibrated to match key characteristics of COVID-19 transmission. An important calibration outcome is the age-dependent fraction of symptomatic cases, with this fraction for children found to be one-fifth of such fraction for adults. We apply the model to compare several intervention strategies, including restrictions on international air travel, case isolation, home quarantine, social distancing with varying levels of compliance, and school closures. School closures are not found to bring decisive benefits unless coupled with high level of social distancing compliance. We report several trade-offs, and an important transition across the levels of social distancing compliance, in the range between 70% and 80% levels, with compliance at the 90% level found to control the disease within 13–14 weeks, when coupled with effective case isolation and international travel restrictions.

[1] Centre for Complex Systems, Faculty of Engineering, University of Sydney, Sydney, NSW 2006, Australia. [2] Marie Bashir Institute for Infectious Diseases and Biosecurity, University of Sydney, Westmead, NSW 2145, Australia. ✉email: mikhail.prokopenko@sydney.edu.au

The coronavirus disease 2019 (COVID-19) pandemic is an ongoing crisis caused by severe acute respiratory syndrome coronavirus 2 (SARS-CoV-2). The first outbreak was detected in December 2019 in Wuhan, the capital of Hubei province, rapidly followed by the rest of Hubei and all other provinces in China. Within mainland China the epidemic was largely controlled by mid- to late March 2020, having generated >81,000 cases (cumulative incidence on 20 March 2020[1]). This was primarily due to intense quarantine and social distancing (SD) measures, including: isolation of detected cases; tracing and management of their close contacts; closures of potential zoonotic sources of SARS-CoV-2; strict traffic restrictions and quarantine on the level of entire provinces (including suspension of public transportation, closures of airports, railway stations and highways within cities); cancellation of mass gathering activities; and other measures aimed to reduce transmission of the infection[2–4].

Despite the unprecedented domestic control measures, COVID-19 was not completely contained and the disease reached other countries. On 31 January 2020, the epidemic was recognised by the World Health Organisation (WHO) as a public health emergency of international concern, and on 11 March 2020, the WHO declared the outbreak a pandemic[5]. Effects of the COVID-19 pandemic have quickly spilled over from the healthcare sector into international trade, tourism, travel, energy and finance sectors, causing profound social and economic ramifications[6]. While worldwide public health emergencies have been declared and mitigated in the past—for example, the "swine flu" pandemic in 2009[7–10]—the scale of socioeconomic disruptions caused by the unfolding COVID-19 pandemic is unparalleled in recent history.

Australia began to experience most of these consequences, with the number of confirmed COVID-19 cases crossing 1000 by 21 March 2020, while (at that time) doubling every 3 days, and the cumulative incidence growth rate averaging 0.20 per day during the first 3 weeks of March 2020 (Appendix A in Supplementary information). In response, the Australian government introduced strict intervention measures in order to prevent the epidemic from continuing along such trends and to curb the devastating growth seen in other COVID-19-affected nations. Nevertheless, there is an ongoing debate on the utility of specific interventions (e.g. school closures), the low compliance with SD measures (e.g. reduction of mass gatherings), and the optimal combination of particular health intervention options balanced against social and economic ramifications, and restrictions on civil liberties. In the context of this debate, there is an urgent requirement for rigorous and unbiased evaluations of available options. The present study makes a contribution towards this requirement and provides timely input into the Australian pandemic response discussion. Specifically, we develop a large-scale Agent-Based Model (ABM) capturing salient features of COVID-19 transmission in Australia, and use it to evaluate the effectiveness of non-pharmaceutical interventions with respect to the population's compliance with the suggested measures.

Governments around the world are presently fighting the spread of COVID-19 within their jurisdictions by developing, applying and adjusting multiple variations on pandemic intervention strategies. While these strategies vary across nations, they share fundamental approaches that are adapted by national healthcare systems, aiming at a broad adoption within societies. In the absence of a COVID-19 vaccine, as pointed out by Ferguson et al.[11], mitigation policies may include case isolation (CI) of patients and home quarantine (HQ) of their household (HH) members, SD of the individuals within specific age groups (e.g. the elderly, defined as >75 years), as well as people with compromised immune systems or other vulnerable groups. In addition, suppression policies may require an extension of CI and HQ with SD of the entire population.

Often, such SD is supplemented by school and university closures.

Our primary objective is an evaluation of several intervention strategies that have been deployed in Australia, or have been considered for a deployment: restriction on international arrivals ("travel ban"); in-home CI of ill individuals; HQ of family members of ill individuals; SD at various population compliance levels up to and including 100%, a full lockdown; school closures (SCs), which affect the behaviour of school children as well as their parents and teachers. We explore these intervention strategies independently and in various combinations, as detailed in "Methods". Each scenario is traced over time and compared to the baseline model in order to quantify its potential to curtail the epidemic in Australia. Our aims are to identify minimal effective levels of SD compliance, and to determine the potential impact of school closures on the effectiveness of intervention measures.

Stochastic ABMs have been established as robust tools for tracing the fine-grained effect of heterogeneous intervention policies in diverse epidemic and pandemic settings[7,8,12–18], including for policy advice currently in place in the USA and the UK[11]. In this study, we follow the ABM approach to quantitatively evaluate and compare several mitigation and suppression measures, using a high-resolution individual-based computational model calibrated to key characteristics of COVID-19 pandemics. The approach uses a modified and extended agent-based model, ACEMod (Australian Census-based Epidemic Model), previously developed and validated for simulations of pandemic influenza in Australia[19–22]. The epidemiological component, AMTraC-19, is developed and calibrated specifically to COVID-19 via reported invariants (outputs) such as the growth rate above. Importantly, our sensitivity analysis shows that key epidemiological outputs from our model (e.g. the growth rate, $R_0$, generation time, etc.) are robust to uncertainty in the input parameters (e.g. the natural history of the disease, fraction of symptomatic cases, etc.).

In investigating possible effects of various intervention policies, we are able to provide clear and tangible goals for the population and government to pursue in order to mitigate the pandemic within Australia. The key result, based on a comparison of several intervention strategies, is an actionable transition across the levels of SD compliance, identified between 70 and 80% levels. A compliance of below 70% is unlikely to succeed for any duration of SD, while a compliance at the 90% level is found to control the disease within 13–14 weeks, when coupled with effective CI, HQ and international travel restrictions. We validate these results by a comparison with the actual epidemic and SD compliance observed in Australia. In doing so, we confirm that the model has successfully predicted the cumulative incidence as well as the timing of both the incidence and prevalence peaks. Moreover, we illustrate trade-offs between these levels and duration of the interventions, and between the interventions' delay and their duration. Specifically, our simulations suggest that a 3-day delay in introducing strict intervention measures lengthens their required duration by over 3 weeks on average, that is, 23.56 days (with standard deviation of 11.167).

## Results

We present results of the high-resolution (individual-based) pandemic modelling in Australia, including a comparative analysis of intervention strategies. As discussed above, we performed our analysis using ACEMod, an established Australian Census calibrated ABM that captures fine-grained demographics and social dynamics[19–22]. The epidemiological component of our model, AMTraC-19, was developed and calibrated to match key characteristics of COVID-19 (see "Methods").

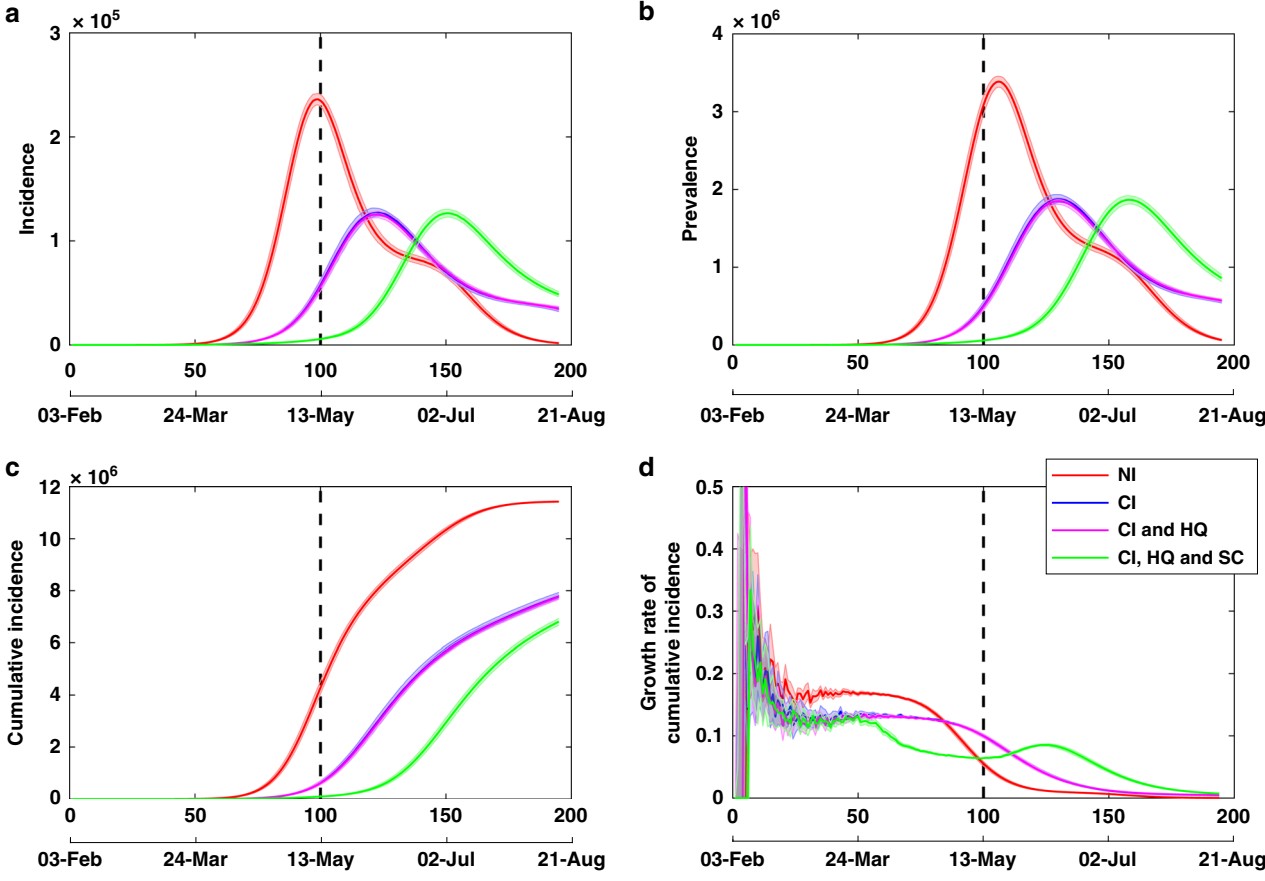

**Fig. 1 Effects of case isolation, home quarantine and school closures.** A combination of the case isolation (CI) and home quarantine (HQ) measures delays epidemic peaks and reduce their magnitude, in comparison to no interventions (NI), whereas school closures (SCs) have short-term effect. Several baseline and intervention scenarios, traced for **a** incidence, **b** prevalence, **c** cumulative incidence and **d** the daily growth rate of cumulative incidence $\dot{C}$, shown as average (solid) and 95% confidence interval (shaded) profiles, over 20 runs. The 95% confidence intervals are constructed from the bias-corrected bootstrap distributions. The strategy with school closures combined with case isolation lasts 49 days (7 weeks), marked by a vertical dashed line. Restrictions on international arrivals are set to last until the end of each scenario. The alignment between simulated days and actual dates may slightly differ across separate runs.

The input parameters were calibrated to generate key characteristics in line with reported epidemiological data on COVID-19. We primarily calibrated by comparing these epidemiological characteristics to the mean of output variables, inferred from Monte Carlo simulations during non-intervention periods, with confidence intervals (CIs) constructed by bootstrapping (i.e. random sampling with replacement) with the bias-corrected percentile method[23].

The key output variables, inferred in concordance with available data, include: a reproductive number $R_0$ of 2.77, 95% CI [2.73, 2.83], $N = 6315$; a generation period $T_{\text{gen}}$ of 7.62 days, 95% CI [7.53, 7.70], $N = 6315$; a growth rate of cumulative incidence during a period of sustained and unmitigated local transmission at $\dot{C} = 0.167$ per day, 95% CI [0.164, 0.170], $N = 20$; and an attack rate in children of $A_c = 6.154\%$, 95% CI [6.15%, 6.16%], $N = 20$. The relatively narrow CIs reflect the intrinsic stochasticity of the simulations carried out for the default values of input parameters. The broad range of possible variations in response to changes in the input parameters, as well as the robustness of the model and its outcomes, is established by the sensitivity analysis (see Appendix D in Supplementary information). This is followed by validation against actual epidemic timeline in Australia (see Appendix H in Supplementary information), confirming that the adopted parametrisation is acceptable.

**Baseline.** A trace of the baseline model—no interventions whatsoever—is shown in Fig. 1, with clear epidemic peaks in both incidence (Fig. 1a) and prevalence (Fig. 1b) evident after 105–110 days from the onset of the disease in Australia, that is, occurring around mid-May 2020. The scale of the impact is very high, with nearly 50% of the Australian population showing symptoms. This baseline scenario is provided only for comparison, in order to evaluate the impact of interventions, most of which were already in place in Australia during the early phase of epidemic growth. To re-iterate, we consider timely intervention scenarios applicable to the situation in Australia at the end of March 2020, with the number of confirmed COVID-19 cases crossing 2000 on 24 March 2020, and the growth rate of cumulative incidence $\dot{C}$ averaging 0.20 per day during the first 3 weeks of March. We observe that the simulated baseline generates $\dot{C} \approx 0.17$ per day, in a good agreement with actual dynamics.

**Case isolation and home quarantine.** All the following interventions include restrictions on international arrivals, triggered by the threshold of 2000 cases. Three mitigation strategies are of immediate interest:

(i) case isolation,

(ii)   in-home quarantine of household contacts of confirmed cases,

(iii)  school closures, combined with (i) and (ii).

These strategies are shown in Fig. 1, with the duration of the SC strategy set as 49 days (7 weeks), starting when the threshold of 2000 cases is reached. The CI strategy coupled with the HQ strategy delays the epidemic peak by ~26 days on average (e.g. shifting the incidence peak from days 97.5 to 123.2, Fig. 1a, and the prevalence peak from days 105 to 130.7, Fig. 1b, on average). In addition, CI combined with HQ reduces the height of the epidemic peak by ~47–49%. The main contributing factor is CI, as adding HQ, with 50% in-home compliance, to CI of 70% symptomatic individuals, delays the epidemic peak by <3 days on average. The overall attack rate resulting from the coupled policy is also reduced in comparison to the baseline scenario (Fig. 1c). However, the CI and HQ strategies, even when coupled together, are not effective for epidemic suppression, with prevalence still peaking in millions of symptomatic cases (1.873M) (Fig. 1b). Such an outcome would have completely overburdened the Australian healthcare system[24].

**School closures**. Adding school closures to the CI and HQ approach also does not achieve a significant reduction in the overall attack rate (Fig. 1). The peaks of both incidence (Fig. 1a) and prevalence (Fig. 1b) are delayed by ~4 weeks (~27 days for both incidence and prevalence). However, their magnitudes remain practically the same, due to a slower growth rate of cumulative incidence (Fig. 1d). This is observed irrespective of the commitment of parents to stay home (Appendix G in Supplementary information). We also traced the dynamics resulting from the SC strategy for two specific age groups: children and individuals >65 years old, shown in Appendix G in Supplementary information. The 4-week delays in occurrence of the peaks are observed across both age groups, suggesting that there is a strong concurrence in the disease spread across these age groups. We also observe that under the SC strategy coupled with CI and HQ, the magnitude of the incidence peak for children increases by ~7% shown in Appendix G in Supplementary information (Supplementary Fig. 9a). This may be explained by increased interactions of children in household and community social mixing environments, when schools are closed. Under this strategy, there is no difference in the magnitude of the incidence peak for the older age group (Appendix G in Supplementary information, Supplementary Fig. 10a). We also note that the considered interventions succeed in reducing a relatively high variance in the incidence fraction of symptomatic older adults, thus reducing the epidemic potential to adversely affect this age group specifically.

In short, the only tangible benefit of school closures, coupled with CI and HQ, is in delaying the epidemic peak by 4 weeks, at the expense of a slight increase in the contribution of children to the incidence peak. While school closures are considered an important part of pandemic influenza response, our results suggest that this strategy is much less effective in the context of COVID-19. The gains are further reduced by other societal costs of school closures, for example, drawing their parents employed in healthcare and other critical infrastructure away from work. There is, nevertheless, one more possible benefit of school closures, discussed in the context of the population-wide SD in Appendix G in Supplementary information.

**Social distancing**. Next, we examine the effects of population-wide SD in combination with CI and restrictions on international arrivals. Here, we present the effects of different compliance levels on the epidemic dynamics. Low compliance levels, set at <70%,

did not show any potential to suppress the disease in the considered time horizon (28 weeks), while the total lockdown, that is, complete SD at 100%, managed to reduce the incidence and prevalence to zero, after 49 days of the mitigation. However, because it is unrealistic to expect 100% compliance in the Australian context, we focus on the practically achievable compliance levels: 70, 80 and 90%, with their duration set to 91 days (13 weeks), shown in Fig. 2.

Importantly, during the time period that the SD level is maintained at 70%, the disease is not controlled, with the numbers of new infected cases (incidence) remaining in hundreds, and the number of active cases (prevalence) remaining in thousands. Thus, 70% compliance is inadequate for reducing the effective reproductive number below 1.0. In contrast, the two higher levels of SD, 80 and 90%, are more effective at suppressing both prevalence and incidence during the 13-week SD period. Figure 2 contrasts these three levels of SD compliance, "zooming in" into the key time period, immediately following the introduction of SD. Crucially, there is a qualitative difference between the lower levels of SD compliance (70%, or less) and the higher levels (80%, or more). For the SD compliance set at 80 and 90%, we observe a reduction in both incidence (Fig. 2a) and prevalence (Fig. 2b), lasting for the duration of the strategy (91 days). With SD compliance of 80%, the disease is not completely eliminated, but incidence is reduced to <100 new cases per day, with prevalence below 1000 by the end of the suppression period (Fig. 2b). It is important to note that while the disease is suppressed during the period over which SD is in effect, resurgence of transmission is likely unless complete or near-complete elimination has been achieved upon cessation of SD measures. Our results suggest that this level of compliance would succeed in eliminating the disease in Australia if the strategy was implemented for a longer period, for example, another 4–6 weeks.

The 90% SD compliance practically controls the disease, bringing both incidence and prevalence to very low numbers of isolated cases (and reducing the effective reproductive number to nearly zero). It is possible for the epidemic to spring back to significant levels even under this level of compliance, as the remaining sporadic cases indicate a potential for endemic conditions. We do not quantify these subsequent waves, as they develop beyond the immediately relevant time horizon. Nevertheless, we do share the concerns expressed by the Imperial College COVID-19 Response Team: "The more successful a strategy is at temporary suppression, the larger the later epidemic is predicted to be in the absence of vaccination, due to lesser build-up of herd immunity"[11]. Given that the herd immunity threshold is determined by $1 - 1/R_0$[25], the extent required to build up collective immunity for COVID-19, assuming $R_0 = 2.77$, may be estimated as 0.64, that is, 64% of the population becoming infected or eventually immunised.

The cumulative incidence for the best achievable scenario (90% SD compliance coupled with CI, HQ, and restrictions on international arrivals) settles in the range of 8000–10,000 cases during the suppression period, with resurgence still possible at some point after intervention measures are relaxed (Fig. 2c). The range of cumulative incidence at the end of the suppression is 8313–10,090 over 20 runs, with the mean of 9122 cases and 95% CI [8898, 9354], constructed from the bias-corrected bootstrap distribution (see Source data file). In terms of case numbers, this is an outcome several orders of magnitude better than the worst-case scenario, developing in the absence of the combined mitigation and suppression strategies.

We compare two sets of scenarios. In our primary scenarios, aligned with the actual epidemic curves in Australia, the SD measures are triggered by 2000 confirmed cases. In alternative

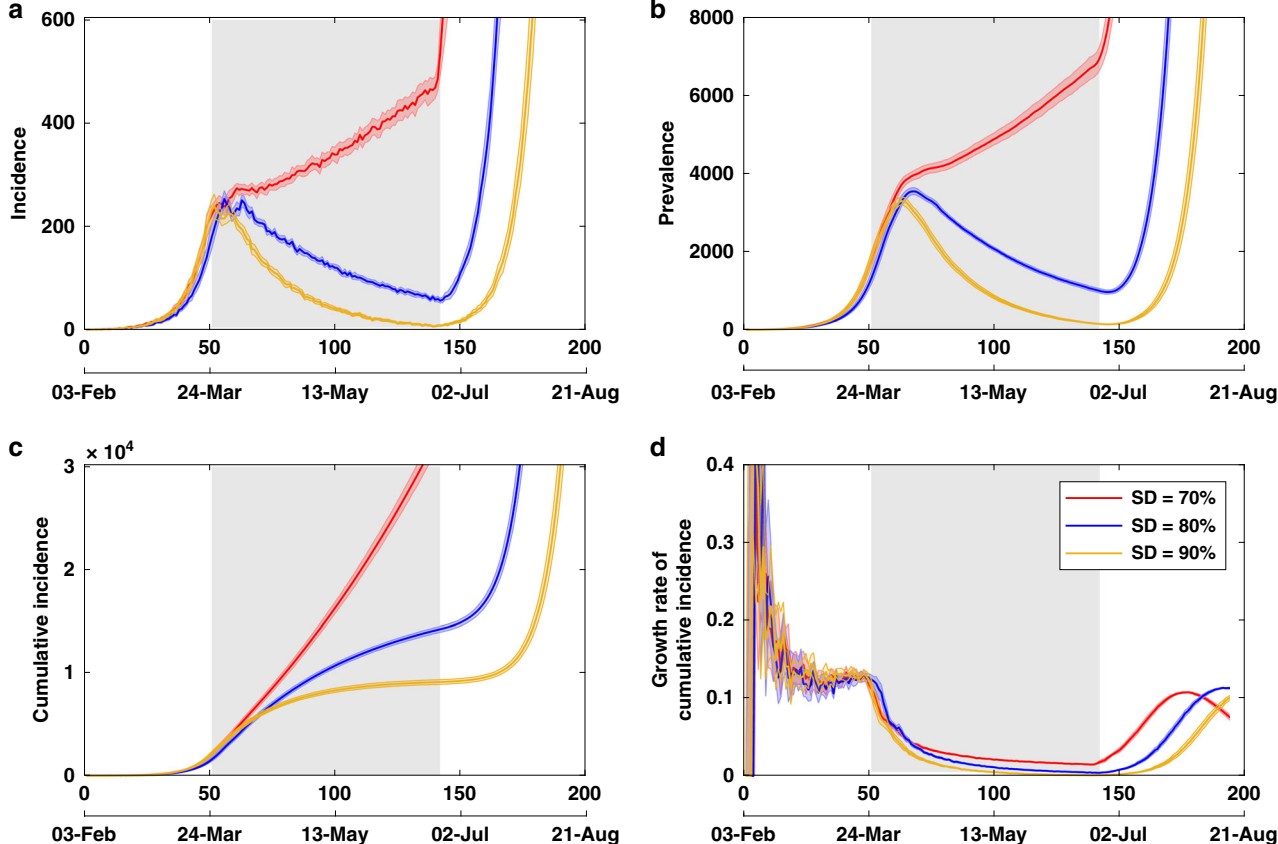

**Fig. 2 Effects of social distancing.** Strong compliance with social distancing (at 80% and above) effectively controls the disease during the suppression period, while lower levels of compliance (at 70% or less) do not succeed for any duration of the suppression. A comparison of social distancing strategies, coupled with case isolation, home quarantine and international travel restrictions, across different compliance levels (70, 80 and 90%). Duration of each social distancing (SD) strategy is set to 91 days (13 weeks), shown as a grey shaded area between days 51 and 142 (the start and end days of SD varied across stochastic runs: for 70% SD the last day of suppression was 141.4 on average; for 80% SD it was 144.2; and for 90% SD it was 141.5, see Source data file). Case isolation, home quarantine and restrictions on international arrivals are set to last until the end of each scenario. Traces include **a** incidence, **b** prevalence, **c** cumulative incidence and **d** the daily growth rate of cumulative incidence $\dot{C}$, shown as average (solid) and 95% confidence interval (shaded) profiles, over 20 runs. The 95% confidence intervals are constructed from the bias-corrected bootstrap distributions. The alignment between simulated days and actual dates may slightly differ across separate runs.

scenarios, the strict suppression measures are initiated earlier, being triggered by crossing the threshold of 1000 cases (Appendix H.1 in Supplementary information). The best agreement between the actual and simulation timelines is found to match a delayed but high (90%) SD compliance, appearing to be followed from 24 March 2020, after a 3-day period with a weaker compliance, which commenced on 21 March 2020 when the international travel restrictions were introduced, as shown in Fig. 3 and detailed in Appendix H.2 in Supplementary information. For the 1000 case threshold scenario, we present the effects of different SD compliance levels (70 and 90%) on the spatial distribution of cases on day 60. These are shown in Appendix I in Supplementary information, as choropleth maps of the four largest Australian Capital Cities: Sydney, Melbourne, Brisbane and Perth.

It is clear that there is a trade-off between the level of SD compliance and the duration of the SD strategy: the higher the compliance, the more quickly incidence is suppressed. Both 80 and 90% compliance levels control the spread within reasonable time periods: 18–19 and 13–14 weeks, respectively. In contrast, lower levels of compliance (at 70% or less) do not succeed for any duration of the imposed SD limits. This quantitative difference is of major policy setting importance, indicating a sharp transition

in the performance of these strategies in the region between 70 and 80%.

Referring to Fig. 4, the identified transition across the levels of compliance with SD may also be interpreted as a tipping point or a phase transition[26]. Various critical phenomena have been discovered previously in the context of epidemic models, often interpreting epidemic diffusion in statistical–mechanical terms, for example, as percolation within a network[27–30]. The transition across the levels of SD compliance is similar to percolation transition in a forest-fire model with immune trees[31]. Distinct epidemic phases are evident in Fig. 4 at a certain percolation threshold between the SD compliance of 70 and 80%, at which the critical regime exhibits the effective reproductive number $R_{\mathrm{eff}} = 1.0$. That is, crossing this regime signifies moving into the phase where the epidemic is controlled, that is, reducing $R_{\mathrm{eff}}$ below 1.0.

We do not attempt to establish a more precise level of required compliance between 70 and 80%. Such a precision would be of lesser practical relevance than the identification of 80% compliance as the minimal acceptable level of SD, with 90% providing a shorter timeframe. The robustness of these results is established by sensitivity analysis presented in Appendix D.2 in Supplementary information.

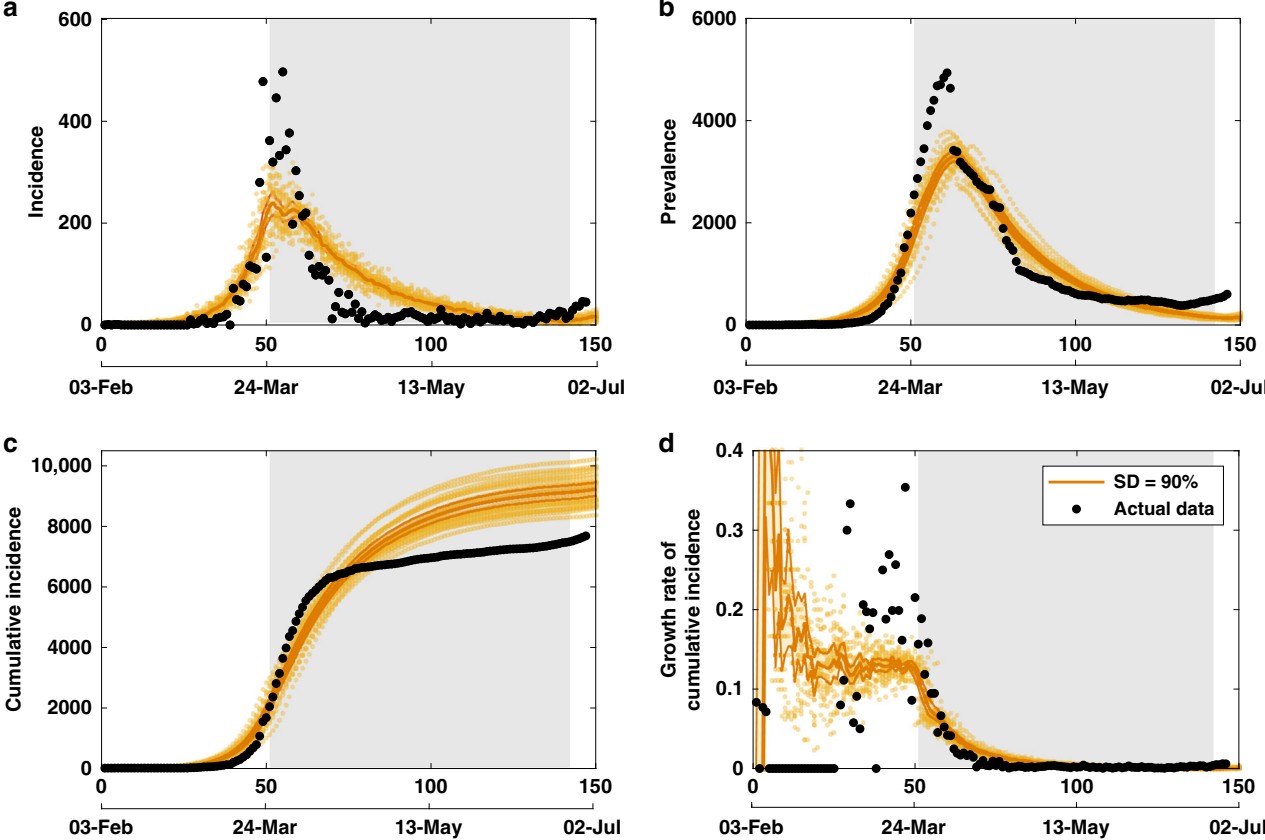

**Fig. 3 Model validation with actual data.** A comparison between actual epidemic curves in Australia (black dots, shown until 28 June 2020), and the primary simulation scenario, using a threshold of 2000 cases (crossed on 24 March 2020) and following 90% of social distancing (SD), coupled with case isolation, home quarantine and international travel restrictions, shown until early July 2020 (yellow colour). Duration of the SD strategy is set to 91 days (13 weeks), shown as a grey shaded area. Case isolation, home quarantine and restrictions on international arrivals are set to last until the end of the scenario. Traces include **a** incidence, **b** prevalence, **c** cumulative incidence and **d** daily growth rate of cumulative incidence, shown as average (solid), 95% confidence interval (thin solid) profiles, as well as the ensemble of 20 runs (scatter). The 95% confidence intervals are constructed from the bias-corrected bootstrap distributions. The alignment between simulated days and actual dates may slightly differ across separate runs. Data sources: refs. [64,67].

In addition, a 3-day delay in introducing strong SD measures is projected to extend the required suppression period by ~3 weeks, beyond the 91-day period considered in the primary scenario (see Appendix H in Supplementary information). Finally, we report fractions of symptomatic cases across mixing contexts (Appendix J in Supplementary information), with the infections through HHs being predominant. Notably, the HH fractions steadily increase with the strengthening of SD compliance, while the corresponding fractions of infections in the workplace and school environments decrease.

**Summary**. In short, the best intervention approach identified in our study is a combination of international travel restrictions, CI, HQ and SD with at least 80%–90% compliance for a duration of ~91 days (13 weeks). These measures have been implemented in Australia to a reasonable degree; however, it is unclear if testing throughput and contact tracing resources are sufficient to facilitate effective interventions if incidence increases substantially. For these reasons, it is our conclusion that SD is likely to continue to be the instrumental line of defense against COVID-19 in Australia. In our study, compliance levels below 80% resulted in higher prevalence at the end of suppression period, and increasing incidence during the SD period.

We point out that our results are relevant only for the duration of the mitigation and suppression, and a resurgence of the disease is possible once these interventions cease, as shown in Fig. 2. We

also note that a rebound in the incidence and prevalence post-suppression period is not unavoidable: more efficient and large-scale testing methods are expected to be developed in several months, and so the resultant contact tracing and CI are likely to prevent a resurgence of the disease. The international travel restrictions are assumed to stay in place. Hence, we do not quantify the precise impact of control measures beyond the selected time horizon (28 weeks), aiming to provide immediately relevant insights. Furthermore, our results should not be seen as policies optimised over all possible parameter combinations, but rather as a clear demonstration of the extent of SD required to reduce incidence and prevalence over 2–6 months.

## Discussion

In this study, we simulated several possible scenarios of COVID-19 pandemic's spread in Australia. The model, AMTraC-19, was calibrated to known pandemic dynamics, and accounted for age-dependent attack rates, a range of reproductive numbers, age-stratified and social context-dependent transmission rates, household clusters (HCs) and other social mixing contexts, symptomatic–asymptomatic distinction, and other relevant epidemiological parameters. An important calibration result was the need for age-dependent fractions of symptomatic agents, with the fraction of symptomatic children found to be one-fifth of that of the adults.

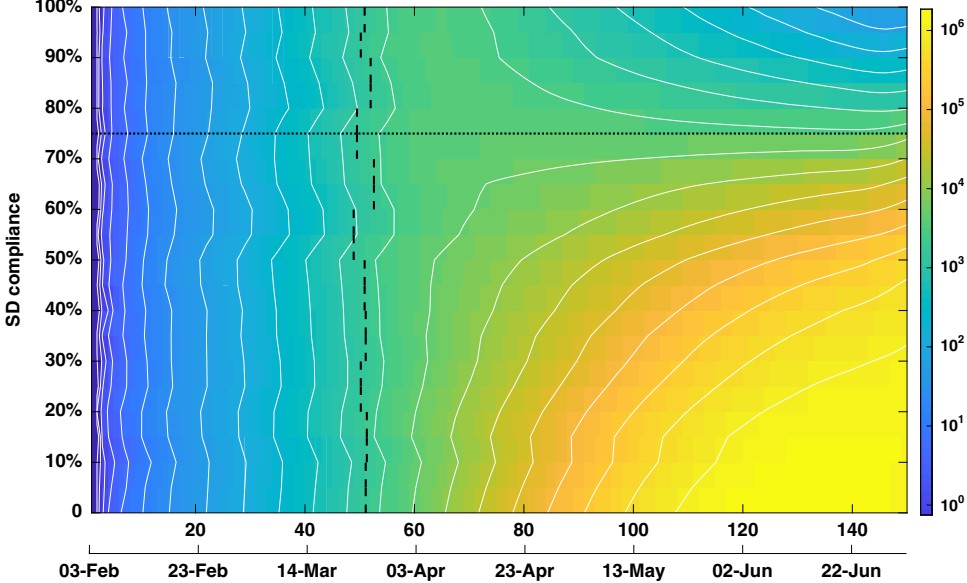

**Fig. 4 Phase transition across the levels of social distancing compliance.** Colour image plot of disease prevalence as a function of time (horizontal axis) and social distancing (SD) compliance (vertical axis). A phase transition is observed between 70 and 80% SD compliance (marked by a dotted line). For SD compliance levels below 80%, the prevalence continues to grow after social distancing is implemented, while for compliance levels at or above 80% the prevalence declines, following a peak formed after ~2 months. The colours correspond to log-prevalence, traced from the epidemic's onset until the end of the suppression period. The isolines trace contours with constant values of log-prevalence. Vertical dashes mark the time when threshold of 2000 is crossed, triggering SD, averaged over 20 runs for each SD level. Social distancing is coupled with case isolation, home quarantine and international travel restrictions. The alignment between simulated days and actual dates may slightly differ across separate runs.

We reported several findings relevant to COVID-19 mitigation and suppression policy setting. The first implication is that the effectiveness of school closures is limited (under our assumptions on the age-dependent symptomatic fractions and the infectivity in children), producing a 4-week delay in epidemic peak, without a significant impact on the magnitude of the peak, in terms of incidence or prevalence. The temporal benefit of this delay may be offset not only by logistical complications, but also by some increases in the fractions of both children and older adults during the period around the incidence peak. As the clinical picture of COVID-19 in children continues to be refined[32], these findings may benefit from a re-evaluation when more extensive paediatric data become available.

The second implication is related to the SD strategy, which showed little benefit for lower levels of compliance (at 70% or less)—these levels do not produce epidemic suppression for any duration of the SD restrictions. Only when the SD compliance levels exceed 80%, there is a reduction in incidence and prevalence. Our modelling results indicate existence of an actionable transition across these strategies between 70 and 80%. In other words, increasing a compliance level just by 10%, from 70 to 80%, may effectively control the spread of COVID-19 in Australia, by reducing the effective reproductive number to near zero (during the suppression period).

We also reported a trade-off between the compliance levels and the duration of SD mitigation, with 90% compliance significantly reducing incidence and prevalence after a shorter period of 91 days (13 weeks). Although a resurgence of the disease is possible once these interventions cease, we believe that this study could facilitate a timely planning of effective intervention and exit strategies. In particular, this study contributed to the report, "Roadmap to Recovery", presented to the Australian Federal Government on 29 April 2020, providing evidence for a comparison between two options. Rather than recommending "a

single dominant option for pandemic response in Australia", the roadmap pointed out considerable and evolving uncertainties, and presented two strategies: (i) a state by state elimination of local community transmissions (with the restrictions remaining for a longer duration, but achieving lower cases and greater public confidence), and (ii) controlled adaptation aimed at some minimal level of symptomatic cases within the health system capacity (with phased and adaptive lifting of restrictions, beginning as early as 15 May 2020, but acknowledging the high likelihood of prolonged global circulation of SARS-CoV-2)[33]. However, a precise evaluation of detailed exit strategies, as well as the probability of elimination, lies outside the scope of our study.

Future research will address several limitations of our study, including a more fine-grained implementation of natural history of the disease, reducing uncertainty around the transmissibility and infectivity in young people, incorporation of more recent Australian Bureau of Statistics (ABS) data from 2020, and an account of hospitalisations and in-hospital transmissions. We also hope to trace specific spatial pathways and patterns of epidemics, in order to enable a detailed understanding of how the infection spreads in diverse circumstances and localities, with the aim to identify the best ways to locate and curtail the pandemic spread in Australia. It would be interesting to contrast our ABM with network-based approaches: while both frameworks depart from the compartmental fully mixed models in capturing specific interactions affecting the infection spread, there are differences in describing the context dependence and ways to intervene[28,34]. In network-based models, the most effective interventions have been found to be those which reduce the diversity of interactions[35], and can be modelled by changes in the topology of contact networks[36]. Thus, one future direction would be a comparison of the epidemic and intervention thresholds across the ABM and network-based models. Other avenues lead to analysis of precursors and critical thresholds for possible emergence of new

strains, as well as various "change points" in the spreading rate[29,37,38], studies of genomic surveillance data interpreted as complex networks[39–41], dynamic models of social behaviour in times of health crises[42–44] and investigations of global socio-economic effects of the COVID-19 pandemic[6,45,46].

## Methods

ACEMod employs a discrete-time and stochastic agent-based model to investigate complex outbreak scenarios across the nation over time. The ACEMod simulator comprises over 24 million software agents, each with attributes of an anonymous individual (e.g. age, gender, occupation, susceptibility and immunity to diseases), as well as contact rates within different social contexts (HHs, HCs, local neighbourhoods, schools, classrooms, workplaces). The set of generated agents captures average characteristics of the real population, for example, ACEMod is calibrated to the Australian Census data (2016) with respect to key demographic statistics. In addition, the ACEMod simulator has integrated layered school attendance data from the Australian Curriculum, Assessment and Reporting Authority, within a realistic and dynamic interaction model, comprising both mobility and human contacts. These social mixing layers represent the demographics of Australia as close as possible to the ABS and other datasets, as described in Appendix F in Supplementary information.

Potential interactions between spatially distributed agents are represented using data on mobility in terms of commuting patterns (work, study and other activities), adjusted to increase precision and fidelity of commute networks[47]. Each simulation scenario runs in 12-h cycles ("day" and "night") over the 196 days (28 weeks) of an epidemic, and agents interact across distinct social mixing groups depending on the cycle, for example, in working groups and/or classrooms during a "day" cycle, and their HHs, HCs and local communities during the "night" cycle. The interactions result in transmission of the disease from infectious to susceptible individuals: given the contact and transmission rates, the simulation computes and updates agents' states over time, starting from initial infections, seeded in international airports around Australia[19,20]. The simulation is implemented in C++11, using the g++ compiler (GCC) 4.9.3 and GNU Autotools (autoconf 2.69, automake 1.15), running under CentOS release 6.9 (upstream Red Hat 4.4.7-18) on a High-Performance Computing service and utilising 4264 cores of computing capacity. Post processing of simulation results is carried out with MATLAB R2020a.

Simulating disease transmission in ACEMod requires both (i) specifics of local transmission dynamics, dependent on individual health characteristics of the agents, such as susceptibility and immunity to disease, driven by their transmission and contact rates across different social contexts; and (ii) a natural disease history model for COVID-19, that is, the infectivity profile from the exposure, to the peak of infectivity, and then to recovery, for a single symptomatic or asymptomatic infected individual. The infectivity of agents is set to exponentially rise and peak at 5 days, after 2 days of zero infectivity. The symptoms are set to last up to 12 days post the infectivity peak, during which time infectiousness linearly decreases to zero. The probability of transmission for asymptomatic/presymptomatic agents is set as 0.3 of that of symptomatic individuals; and the age-dependent fractions of symptomatic cases are set as $\sigma_c = 0.134$ for children, and $\sigma_a = 0.669$ for adults. These parameters were calibrated to available estimates of key transmission characteristics of COVID-19 spread, implemented in AMTraC-19, the Agent-based Model of Transmission and Control of the COVID-19 pandemic in Australia.

**Calibration.** Despite several similarities with influenza, COVID-19 has a number of notable differences, specifically in relation to transmissions across children, its reproductive number $R_0$, incubation and generation periods, proportion of symptomatic to asymptomatic cases, the infectivity of the asymptomatic and presymptomatic individuals and so on (see Appendix B in Supplementary information). While uncertainty around the reproductive number $R_0$, the incubation and generation periods, as well as the age-dependent attack rates of the disease, have been somewhat reduced[3,4,48], there is still an ongoing effort in estimating the extent to which people without symptoms, or exhibiting only mild symptoms, might contribute to the spread of the coronavirus[49]. Furthermore, the question whether the ratio of symptomatic to total cases is constant across age groups, especially children, has not been explored in studies to date, remaining another critical unknown.

Thus, our first technical objective was to calibrate the AMTraC-19 model for specifics of COVID-19 pandemic, in order to determine key disease transmission parameters of AMTraC-19, so that the resultant dynamics concur with known estimates. In particular, we investigated a range of the reproductive number $R_0$ (the number of secondary cases arising from a typical primary case early in the epidemic). The range 2.0–2.5 has been initially reported by the WHO-China Joint Mission on Coronavirus Disease 2019[3]. Several studies estimated that before travel restrictions were introduced in Wuhan on 23 January 2020, the median daily reproduction number $R_0$ in Wuhan was 2.35, with 95% CI [1.15, 4.77][50]. On 15 April 2020, Australian health authorities reported $R_0$ in the range 2.6–2.7[33], while more recent Australian and international studies investigated $R_0$ in the range 2.5–3.5[24,33,38,44]. For example, a median $R_0 = 3.4$ (CI [2.4, 4.7]) was used in a model of the COVID-19 spread in Germany[38], while the estimates reviewed by Liu et al.[51] ranged from 1.4 to 6.49, with a mean of 3.28 and a median of 2.79. In our

**Table 1 The reproductive number $R_0$ and the generation period $T_{gen}$ (with 95% confidence intervals (CIs), constructed from the bias-corrected bootstrap distribution), for various values of the scaling parameter $\kappa$.**

| $\kappa$ | $R_0$ | 95% CI | $T_{gen}$ | 95% CI | Sample size |
|---|---|---|---|---|---|
| 2.00 | 1.94 | [1.91, 1.98] | 6.92 | [6.81, 7.02] | 6274 |
| 2.25 | 2.39 | [2.35, 2.44] | 7.36 | [7.27, 7.45] | 6372 |
| 2.50 | 2.59 | [2.54, 2.64] | 7.46 | [7.37, 7.55] | 6351 |
| 2.75 | 2.77 | [2.73, 2.83] | 7.62 | [7.53, 7.70] | 6315 |
| 3.00 | 3.12 | [3.10, 3.21] | 7.74 | [7.66, 7.82] | 6413 |

model, $R_0$, our output variable, $y_1$, was investigated between 1.94 and 3.12, see Table 1, by varying a scaling factor $\kappa$ responsible for setting the contagiousness of the simulated epidemic, as explained in Appendix C in Supplementary information[19,21].

We aimed for the generation period $T_{gen}$, that is, our output variable $y_2$, to stay in the range 6.0–10.0[18,52,53]. This is also in line with the reported mean serial interval of 7.5 days (with 95% CI [5.3, 19])[52].

In addition, we aimed to keep the resultant daily growth rate of cumulative incidence $\dot{C}$, output variable $y_3$, ~0.2 per day, in order to be consistent with the disease dynamics reported in Australia and internationally (see Appendix A in Supplementary information). Our focus was to characterise the rate of a rapid infection increase during the sustained but unmitigated local transmission. This calibration target was chosen at the time, mid-March 2020, to complement $R_0$ and the generation period, given the lack of data on the epidemic peak values, and fragmented patient recovery and prevalence data. By that time, despite different initial conditions and disease surveillance regimes, as well as diversity of case definitions, several countries exhibited a similar growth pattern. This suggested that a steady growth rate of ~0.2 per day may provide a consistent calibration target during the early growth period, with seven out of the top eight affected nations settling around this rate after a noisy transient (except South Korea where the initial growth had the cluster nature, following a superspreading event[54]).

Another key constraint was a low attack rate in children, $A_c$, that is, our output variable $y_4$, reported to be in single digits. For example, only 2.4% of all reported cases in China were children, while a study in Japan observed that "it is remarkable that there are very few child cases aged from 0 to 19 years", with only 3.4% of all cases in this age group[55].

The calibration was aimed at satisfying our key constraints, given by the expected ranges of output variables. In doing so, we varied several "free" parameters, such as transmission and contact rates, the fraction of symptomatic cases (making it age-dependent), the probability of transmission for both symptomatic and asymptomatic agents, and the infectivity profile from the exposure. Specifically, we explored the time to infectivity peak, our input parameter $x_1$, in proximity to known estimates of the mean incubation period, that is, between 4 and 7 days, calibrating the time to peak to 5.0 days. In several studies, the mean incubation period was reported as 5.2 days, 95% CI [4.1, 7.0][52], while being distributed around a mean of ~5 days within the range of 2–14 days with 95% CI[56]. We also varied the symptoms' duration after the peak of infectivity, that is, recovery period, our input parameter $x_2$, between 7 and 21 days, and calibrated it at 12.0 days, on a linearly decreasing profile from the peak.

The contact and transmission rates across various mixing contexts detailed in Appendices C and E in Supplementary information. The probability of transmission for asymptomatic/presymptomatic agents, our input parameter $x_3$, was set as 0.3 of that of symptomatic individuals (lower than in the ACEMod influenza model), having been explored between 0.05 and 0.45. Both symptomatic and asymptomatic infectivity profiles were changed to increase exponentially after a latent period of 2 days, reaching the infectivity peak after 5 days, with the onset of symptoms distributed across agents during this period, see Appendix C in Supplementary information.

The fraction of symptomatic cases, our input parameter $x_4$, was investigated between 0.5 and 0.8, and set to two-thirds of the total cases ($\sigma_a = 0.669$), which concurs with several studies. For example, the initial data on 565 Japanese citizens evacuated from Wuhan, China, who were symptom-screened and tested, indicated that 41.6% were asymptomatic, with a lower bound estimated as 33.3% (95% CI [8.3, 58.3])[57]. The proportion of asymptomatic cases on the Diamond Princess cruise ship was estimated between 17.9 (95% credible interval (CrI): 15.5–20.2%) and 39.9% (95% CrI: 35.7–44.1%)[58], noting that most of the passengers were 60 years and older, and more likely to experience more symptoms. The modelling study of Ferguson et al.[11] also set the fraction of symptomatic cases to $\sigma = 0.669$.

However, we found that our output variables were within the expected ranges only when this fraction is age-dependent, with the fraction of symptomatic cases among children, our input parameter $x_5$, calibrated to one-fifth of the one for adults, that is, $\sigma_c = 0.134$ for children, and $\sigma_a = 0.669$ for adults. This calibration outcome per se, achieved after exploring the range $\sigma_c \in [0.05, 0.25]$, is in agreement with the reported low symptomaticity in children worldwide, and the observation

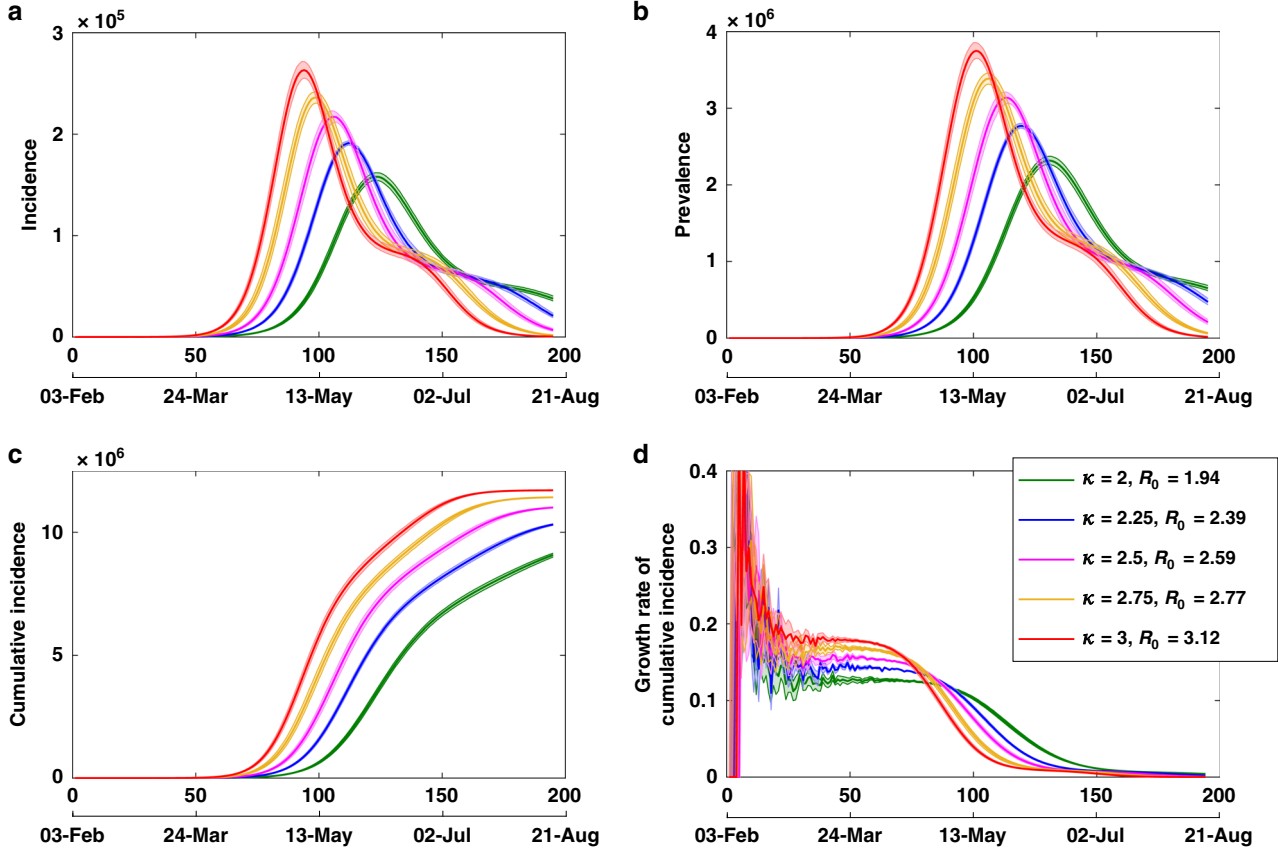

**Fig. 5 Model calibration with scaling factor κ.** Tracing **d** the expected growth rate of cumulative incidence $\dot{C}$ per day, while varying scaling factor κ (proportional to the reproductive number $R_0$), with **a** incidence, **b** prevalence and **c** cumulative incidence. Averages over 20 runs are shown as solid profiles, with 95% confidence intervals shown as shaded profiles. The 95% confidence intervals are constructed from the bias-corrected bootstrap distributions. The alignment between simulated days and actual dates may slightly differ across separate runs.

that "children are at similar risk of infection as the general population, although less likely to have severe symptoms"[59]. Another study of epidemiological characteristics of 2143 paediatric patients in China noted that over 90% of patients were asymptomatic, mild or moderate cases[60].

In summary, this combination of parameters resulted in the dynamics that matched several COVID-19 pandemic characteristics. It produced the following estimates and their CIs, constructed from the bias-corrected bootstrap distribution:

- the reproductive number $R_0 = 2.77$, with 95% CI [2.73, 2.83] (sample size $N = 6315$);
- the generation period $T_{gen} = 7.62$ days, with 95% CI [7.53, 7.70] ($N = 6315$);
- the growth rate of cumulative incidence, determined at day 50, during a period of sustained unmitigated local transmission, $\dot{C} = 0.167$ per day, with 95% CI [0.164, 0.170] and range 0.156–0.182 ($N = 20$);
- the attack rate in children $A_c = 6.154\%$, with 95% CI [6.15, 6.16%] and range 6.14–6.16% ($N = 20$).

Both the reproductive number and the generation period correspond to κ = 2.75 (see Table 1 for other values of κ). The resultant dynamics are shown in Figs. 5 and 6. The sensitivity analysis of the output variables to changes in the input parameters is presented in Appendix D.1 in Supplementary information. We point out that, in hindsight, one may choose more comprehensive calibration targets and refine the model with different parametrisations. The model presented in this study was calibrated by 24 March 2020, using Australian and international incidence and prevalence data from two preceding months, as well as constraints on the output variables detailed above. At the time, a limited testing capacity resulting in possible under-reporting of cases (especially paediatric) may have introduced a potential bias in model calibration. Nevertheless, the study is described here as an approach, which succeeded in accurately predicting the epidemic peaks in Australia in early April (both incidence and prevalence), while providing timely advice on relevant pandemic interventions.

**Fraction of local community transmissions**. We trace scenarios of COVID-19 pandemic spread in Australia, initiated by passenger arrivals via air traffic from overseas. This process maintains a stream of new infections at each time step, set in proportion to the average daily number of incoming passengers at that airport[20,21].

These infections occur probabilistically, generated by binomial distribution $B(P, N)$, where $P$ and $N$ are selected to generate one new infection within a 50 km radius of the airport, per 0.04% of incoming arrivals on average.

In a separate study[41], we directly compared the fractions of local transmissions detected by our ABM with the genomic sequencing of SARS-CoV-2, carried out in a subpopulation of infected patients within New South Wales, the most populous state of Australia, until 28 March 2020. Only a quarter of sequenced cases was deemed to be locally acquired (cases who had not travelled overseas in the 14 days before illness onset), and this was in concordance with the trace obtained from our ABM model. Specifically, having simulated the 5-week period preceding intervention measures, we inferred all local transmission links within HHs, HCs, and local government areas that map to the census statistical areas (SAs). Each directed link connecting two infected individuals in the same mixing context is detected if the infected agents share the same HH, HC or SA identifier, and the direction is inferred using the relevant simulation time steps. Then, the fraction of local community transmissions is determined as the ratio between the number of the inferred transmission links and the number of total infections during the corresponding time period. These fractions ranged between 18.6% (std. dev. 2.9%) for HH and HC combined, and 34.9% (std. dev. 8.2%) for all transmissions within HH, HC and SA, broadly agreeing with the fraction identified through genomic surveillance: 25.8% for all local transmissions[41].

**Sensitivity analysis**. We performed our sensitivity analysis using the local (point-based) sensitivity analysis (LSA)[61], as well as global sensitivity analysis with the Morris method (the elementary effect method)[62]. Each method computes the response of an "output" variable of interest, for example, the generation period, to the change in an "input" parameter, for example, the fraction of symptomatic cases. The response $F_{i,j}$ of the state variable $y_j$ to parameter $x_i$ from a scaled vector of all $k$ input parameters, $\mathbf{X} = [0, 1]^k$, is determined as a finite difference

$$F_{i,j} = \frac{y_j(x_1, x_2, \ldots, x_i + \Delta, x_{i+1}, \ldots, x_k) - y_j(\mathbf{X})}{\Delta},$$ (1)

where Δ is a discretisation step, dividing each dimension of the parameter space. The distribution of each response $F_{i,j}$ is obtained by repeated random sampling with a number of simulation runs per step. In LSA, an input parameter is varied, while keeping other inputs set at their base points, that is, default values. In the

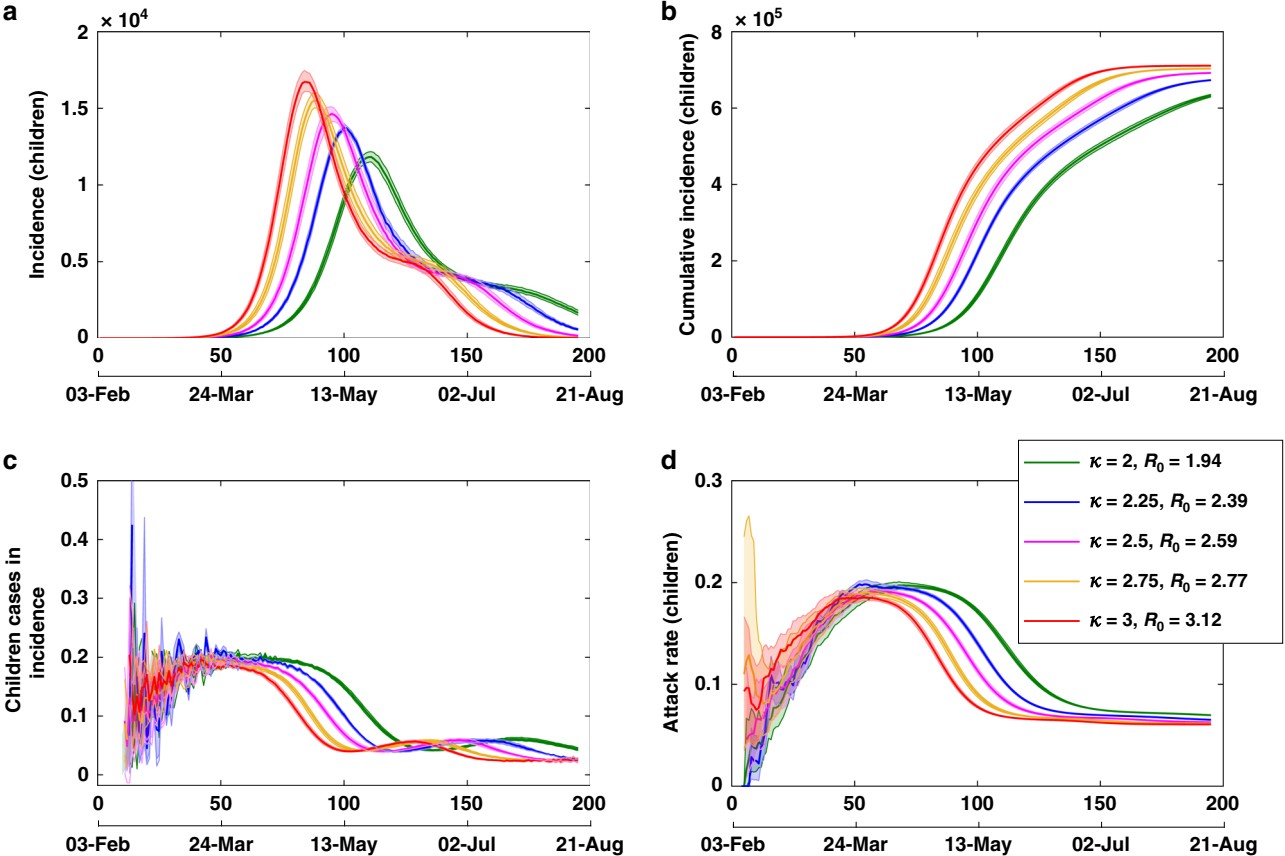

**Fig. 6 Model calibration: epidemic curves for children.** Tracing **d** the attack rate in children, while varying scaling factor κ (i.e. reproductive number $R_0$), with **a** incidence, **b** cumulative incidence and **c** incidence fraction for children. Averages over 20 runs are shown as solid profiles, with 95% confidence intervals shown as shaded profiles. The 95% confidence intervals are constructed from the bias-corrected bootstrap distributions. The alignment between simulated days and actual dates may slightly differ across separate runs.

Morris method, an input parameter is varied at a number of different points sampled across the domains of other parameters. The mean $\mu_{i,j}^*$ of the absolute response $|F_{i,j}|$ serves to capture the influence of the parameter $x_i$ on the output $y_j$: a large mean suggests a higher sensitivity. The standard deviation $\sigma_{i,j}$ of the response $F_{i,j}$ is a complementary measure of sensitivity: a large deviation indicates that the dependency between the input and output is nonlinear. In the Morris method, a large deviation may also indicate that the input parameter interacts with other parameters[63]. Importantly, the responses are not directly comparable across the output variables, and instead are ranked across the inputs for each output. A model is generally considered robust if most of the dependencies are characterised by low means and deviations, with the variations contained within acceptable ranges of the output variables. Appendix D in Supplementary information summarises the investigated ranges and results of the sensitivity analysis.

### Intervention strategies

*International travel restrictions.* In our model, restriction on international arrivals is set to be enforced from the moment when the number of confirmed infections exceeds the threshold of 2000 cases. This concurs well with the actual epidemic timeline in Australia, which imposed a ban on all arrivals of non-residents, non-Australian citizens, from 9 p.m. of 20 March 2020, with a requirement for strict self-isolation of returning citizens. The number of COVID-19 cases crossed 1000 cases on 21 March 2020, and doubled to slightly over 2000 on 24 March 2020, so the 2000 threshold chosen on our model reflects a delay in implementing the measures. The restriction on international arrivals is included in modelling of all other strategies, and is not traced independently, as this mitigation approach is not under debate.

*Case isolation.* The CI mitigation strategy assumes that 70% of symptomatic cases stay at home, reduce their non-household contacts by 75% (so that their transmission rates decrease to 25% of the baseline rate) and maintain their household contacts (i.e. their transmission rates within household remain unchanged). The assumption that even relatively mild symptomatic cases are identified and isolated is justified by the practice adopted in Australia where a comprehensive disease surveillance regime was consistently implemented. This included screening of

syndromic fever and cough in combination with exhaustive case identification and management, thus enabling early detection (e.g. >1% of the Australian population has been tested for the coronavirus by early April 2020, and the numbers of tests conducted in Australia per new confirmed case of COVID-19, as well as per capita, remain among the highest in the world)[24,41,64,65].

*Home quarantine.* In our model of the HQ strategy for household contacts of index cases, we allow compliance to vary within affected households (i.e. at the individual level). In our implementation, 50% of individuals will comply with HQ if a member of their household becomes ill. We simulate this as a reduction to 25% of their usual non-household contact rates, and a consequent doubling of their contact rates within the household. Both CI and HQ strategies are assumed to be in force from the first day of the epidemic, as has been the case in Australia.

*Social distancing.* If an individual complies with SD, all working group contacts are removed, and all non-household contact rates are set to 50% of the baseline value, while keeping contact rates within households unaltered. To simulate imposition of the intervention policy by the federal government, the SD strategy is triggered by crossing the threshold of 2000 cases (matching the actual timeline on 24 March 2020). An alternative threshold of 1000 cases, matching the actual numbers reported on 21 March 2020, is considered to evaluate a delayed introduction of strong SD measures (Appendix H in Supplementary information). In our study, we vary the SD compliance level from 0 to 100% (full lockdown); the compliance level is simply the percentage of individuals who comply with the measure.

*School closures.* School closure removes students, their teachers and a fraction of parents from daytime interactions (their corresponding transmission rates are set to zero), but increases their interaction rates within households (with a 50% increase in household contact rates). All students and teachers are affected. For each affected household, a randomly selected parent chooses to stay at home, with a varying degree of commitment. Specifically, we compared 25 or 50% commitment, as in Australia there is no legal age for leaving school-age children home alone for a reasonable time, in relevant circumstances. This parameter range is concordant with the report of ABS, summarising a survey of household impacts of

**Table 2 The micro- and macro-distancing parameters: macro-compliance levels and context-dependent micro-distancing levels.**

| Strategy | Macro-distancing | Micro-distancing contacts | | |
|---|---|---|---|---|
| | Compliance levels | Household | Community | Workplace/school |
| No intervention | 100% | 100% | 100% | 100% |
| Case isolation | 70% | 100% | 25% | 25% |
| Home quarantine | 50% | 200% | 25% | 25% |
| School closure (children) | 100% | 150% | 150% | 0% |
| School closure (parents) | 25 or 50% | 150% | 150% | 0% |
| Social distancing | 0–100% | 100% | 50% | 0% |

COVID-19 during early April 2020: the proportion of adults keeping their children home from school or childcare reached 24.9%[66]. The upper considered limit, a half of parents, accounts for reasonable scenarios ensuring adequate parental supervision. School closures are assumed to be followed with 100% compliance, and may be concurrent with all other strategies described above. The SC strategy is also triggered by crossing the threshold of 2000 cases. We note that the Australian Federal Government has, so far, not enforced schools closures, and so we investigate the SC intervention separately from, or coupled with, the SD strategy. Hence, the evaluation of school closures provides an input to policy setting, rather than forecasts possible epidemic dynamics.

*Compliance.* The agents affected by various compliance choices are determined in the beginning of each simulation run, with dependency between voluntary measures that does not allow an individual to be compliant with HQ if they are not also compliant with CI. Then, the relevant changes in contact behaviour are applied to the selected agents in every 12-h cycle. The restrictions are applied in a specific order: CI, HQ, SD and SC, with only the most relevant distancing assigned during each simulation cycle. For example, if a student is ill and in CI, the contact reduction factors associated with home quarantine, SD, and school closure would not apply to them, even if they are considered compliant with those measures. The micro- and macro-distancing parameters defining the levels of compliance, together with the affected non-household and household contacts are summarised in Table 2.

*Duration of measures.* While the CI and HQ strategies are assumed to last during the full course of the epidemic, we vary the duration of SD and/or SC strategies across a range of intervals, with a specific focus on 49 and 91 days, that is, 7 or 13 weeks.

**Reporting summary**. Further information on research design is available in the Nature Research Reporting Summary linked to this article.

## Data availability
The data can be made available to approved bona fide researchers after their host institution has signed a Data Access/Confidentiality Agreement with the University of Sydney. Mediated access will enable data to be shared and results to be confirmed without unduly compromising the University's ability to commercialise the software. To the extent that this data sharing does not violate the commercialisation and licensing agreements entered into by the University of Sydney, the data will be made publicly available after the appropriate licensing terms agreed. Post-processing Source Data and Supplementary Data (Supplementary Data 1 and 2) are provided with this paper. Source data are provided with this paper.

## Code availability
The code can be made available to approved bona fide researchers after their host institution has signed a Data Access/Confidentiality Agreement with the University of Sydney. Mediated access will enable code to be shared and results to be confirmed without unduly compromising the University's ability to commercialise the software. To the extent that this code sharing does not violate the commercialisation and licensing agreements entered into by the University of Sydney, the code will be made publicly available after the appropriate licensing terms agreed.

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

## Acknowledgements

We are grateful to Stuart Kauffman, Edward Holmes, Joel C. Miller, Paul Ormerod, Kristopher Fair, Philippa Pattison, Mahendra Piraveenan, Manoj Gambhir, Joseph Lizier, Peter Wang, John Parslow, Jonathan Nolan, Neil Davey, Vitali Sintchenko, Tania Sorrell, Ben Marais, and Stephen Leeder, for discussions of various intricacies involved in agent-based modelling of infectious diseases, and computational epidemiology in general. We were supported through the Australian Research Council grants DP160102742 (S.L.C., N.H., O.M.C., C.Z., M.P.) and DP200103005 (M.P.). ACEMod is registered under The University of Sydney's invention disclosure CDIP Ref. 2019-123. AMTraC-19 is registered under The University of Sydney's invention disclosure CDIP Ref. 2020-018. We are thankful for a support provided by High-Performance Computing (HPC) service (Artemis) at the University of Sydney.

## Author contributions

S.L.C., N.H., O.M.C. and M.P. developed and calibrated COVID-19 epidemiological model. C.Z. implemented intervention strategies. S.L.C. carried out computational simulation, prepared figures, source and supplementary data files. S.L.C., C.Z., O.M.C. and M.P. performed sensitivity analysis and tested the model. M.P. conceived the study and drafted the manuscript, with all authors contributing. All authors contributed to analysis and interpretation of the results, and gave final approval for publication.

## Competing interests

The authors declare no competing interests.
