## [Peer Review File · Nature Communications]

REVIEWER COMMENTS

Reviewer #2 (Remarks to the Author):

This paper is clearly topical, and clearly likely to have impact (indeed, I know that this work has already had impact in the Australian media and through advice to government). The authors adapt an existing epidemic model, developed previously by them, and apply it to the current coronavirus pandemic. The model incorporates detailed demographic information and is therefore tuned to the Australia context, but there are obviously implications for other similar nations. To adapt this model they need to carefully tune parameters and calibrate their model by ensuring that it provides appropriate age stratification. The model is thoroughly described and their work with the model is a carefully constructed experiment (which is the primary original contribution to Nature Communications) which allows the authors to infer the effectiveness of various control measures. I am in favour of publication.

Comment/Suggestions:

- statements in the introduction about 80-90% compliance imply a confidence interval. How? Similarly, the figure "almost four weeks on average" should be quantified with some statement of confidence.
- one of the main conclusions of this paper is the effectiveness of "social" distancing. A simplistic way to model this would be to change the topology of the contact network (for example: <https://arxiv.org/abs/2004.10396> - for obvious reasons, I don't insist on citation, but I am interested in the authors opinion of the contrast). Some commentary on how the demographic method compares to network based approaches (which would imply the exponential growth rate becoming linear) would be interesting.
- this work is also now (as I write this) becoming doubly relevant in the Australian context as the government considers removing restrictions. Some discussion on the implications of this model for those scenario planning exercises would be useful. For example, which measures should be reversed and (potentially) in which order? Which have most "risk"?
- R_0 figures are given to 3 significant figures, and elsewhere data is reported up to 7 significant digits. This seems to be excessive, given the nature of the work and the assumptions driving the model.
- throughout, figures are illustrated with standard deviations, and from 10 simulations. However, I suspect that the distribution is skew and a (perhaps) 90% confidence interval might be more appropriate. Moreover, it is unclear to me that 10 simulations really is a reasonable expectation. I would've thought the computation would be achievable with much larger numbers of replicates.
- In addition to the previous point, simulations throughout seem to be reported with very tight uncertainty estimates. If I understand it correctly, that uncertainty is due to random outcomes in the agent based simulation - and not parameter uncertainty. Given that the 90% confidence intervals of most of the key parameters is huge (R_0 in (1.15,4.77) in one study), some further comment, or computation indicating the sensitivity of the simulation to these parameters (and hopefully it's robustness) would be useful.
- Fig. 2(a) seems to indicate an uptick in cases before the end of control - or immediately after. Given the 1-2 week latency period I find both these results rather surprising.
- The quote concerning the Imperial College result needs some further qualification - what is the expected impact on the total number of infections?
- It would be nice if Fig. 3 could be extended with contours, and more than 10 simulations
- In the case of Australia (and a small number of other territories) the pertinent question is not what will we need to do to prevent/mitigate a "second wave", but rather what is the probability of elimination (of the virus). Comment.
- ABS is not defined on first usage (page 11)
- conclusions on school closure depend (I assume) on rather uncertain assumptions about the transmissibility and infectivity in young people. It would be good to better describe these limitations of the stated results.
- page 19: What is X_i ? Do the authors mean X_i ? Similarly equation (2) is not defined.

- I find the demarcation of Adult (>18) and Child (<19) to be somewhat irritating - what about someone who is 18.5!? Perhaps and \geq sign could be used!? I realise (I assume) that the model assigns age to be integer valued, but still....
- What is Citymapper?
- Table 4 doesn't make sense to me. Some of the of the numbers (percentages, not fractions) go up when I would expect them to go down. Perhaps I am missing something that could maybe be explained better - but please check the data.
- Similarly, given that SC has been a clear and controversial part of the Australian response, it is unexpected that it only features in one scenario.

Overall, the paper could do with more careful proof-reading. There are phrases that are unclear or, at least, could be clearer. Examples include, but are not limited to:

- the statement in the abstract about calibration is unclear on whether this is a property fitted to the data or an observed effect of the model confirming expected data
- the second sentence has 5 commas
- second paragraph "incidence, incidence" - do you mean "incidence" and "cumulative incidence"?
- Sec. 2., second paragraph: It is unclear what the example is an example of
- "the curtailing the epidemic"
- page 5: "case isolation and home quarantine alone" - does this mean alone and separately or together but exclusive of any other control?
- page 8: what is "regime passing"

Michael Small

Reviewer #3 (Remarks to the Author):

This paper uses a detailed agent-based model (ACEMOD) of COVID-19 transmission to investigate the use of control strategies against COVID-19 in the Australian context. The model is spatially explicit and uses various national datasets, including census and school-attendance data to establish social mixing layers including schools, households, workplaces etc. and has adapted from previous studies looking at pandemic influenza.

The authors look at the specific interventions of case isolation, contact quarantine, school closure and social distancing with various levels of compliance. They find a transition in epidemic outcomes with social distancing compliance from 70-90%, with 70% insufficient to stop epidemic growth and 90% providing the potential for local elimination.

The paper is of interest because it applies a model that facilitates individual-level assessment of interventions that affect mobility to a realistic context. If the conclusions from this study could be considered generalisable, this would be of high interest both in the Australian and international contexts.

In assessing this generalisability, I will focus on what I consider to be the critical aspects of applied epidemiological modelling in this context:

- (i) the degree to which assumptions regarding relevant parameters underpinning the model are reasonable
- (ii) the degree to which the model reflects available epidemic data through the calibration process
- (iii) the degree to which conclusions about interventions can inform policy.

In relation to (i), the natural history parameters for the model seem reasonable. I suspect that transmissibility is in reality somewhat more peaked around the onset of symptoms (and declines more quickly afterwards) but I don't believe there are robust data as yet to confirm this.

In relation to asymptomatic cases - their relative infectivity is currently unknown (and really the definition of "asymptomatic" in the literature is a woolly concept at present). I think the assumed asymptomatic fraction is reasonable - there is probably more of a gradient of this with age but the step-function is an ok approximation here. The generation interval for simulations here (~ 6.4 days) is similar to the earlier data from China and the assumptions used in the imperial college reports - there is considerable debate about this value, with a number of studies finding values in the range 4-5 days. However, these have tended to reflect settings in which mitigation is occurring which can be expected to contract the generation interval. So I feel this is reasonable. The values of R_0 used here are in the range 1.6 to 2.8, which cover a roughly appropriate range.

The other element requiring parameterisation are the interventions. Part of the trouble here is that the main text does not necessarily highlight the relevant - e.g. the description of seeding is buried at the bottom of appendix section E and indicates the 0.04% of arrivals generate a new infection. In truth this was non-constant and growing from a much lower value initial to likely significantly above this by the time travel restrictions were initiated - how does this compare with the actual data on importations?

In relation to isolation and quarantine - I could not find a clear description of when isolation occurred. Is this the day that symptoms occur? This would be unrealistic on average particularly for mild cases. I also think that basing the assumptions on the Imperial model is mistaken - in the context of the UK, wide-spread testing was not available. I think while those assumptions might be reasonable if your diagnosis is unknown, both cases and contacts would be expected to more strongly alter their behaviour in response to a known diagnosis.

The assumptions about parents staying home also seem unrealistic for school closure. While there may be no set legal age at which children must be supervised at home, it would not be considered reasonable to leave primary school age-children home alone, particularly in circumstance where they are supposed to have supervised learning. I may have mis-interpreted the parameterisation here but it really needs to be spelled out clearly and to be more clearly justified in terms of choice.

In relation to changes in contact behaviour - are these percentages calculated on a daily basis (rather than a 12-hour cycle basis?). Would be good to clarify that.

In regard to social distancing - it concerns me that this is handled via a single parameter. Surely here was an opportunity to look at how compliance, together with the reduction of non-household contacts (and potentially even household contacts) contributed? Australian guidance related to both micro and macro distancing with visitation of other households etc. excluded for a significant period.

Also, in regard to combination strategies - are there interactions between the contact modifications that need to be considered when combining SC and social distancing. It seems that there is potential for rates of transmission to rise under school closure combined with social distancing and while possible, I'm not sure that actually reflects reality.

In relation to (ii) calibration - this is always a challenge for agent-based models. I have three main comments here

(i) Calibration is a critical part of a reader's acceptance of the model. The fact that there are no calibration results presented in the main text is a big issue for me. Figures 5 and 6 do not present calibration results ... they present intermediate results that can roughly be compared against data that is not shown in the figures. Ideally at the start of the results there would be a comprehensive comparison of the model to the data used in informing it.

(ii) I am unconvinced by the data used to roughly calibrate growth rates. Firstly, cumulative incidence is a poor measure - this is inherently biased by the initial observations, which are almost always inaccurate because surveillance plays catch-up with the epidemic. This concern can be

applied to Italy, the UK and the US and to a lesser extent Germany. South Korea is not a good example because of the cluster nature of the initial growth and Iran's surveillance has many fundamental problems that mean it is not a good example for measurement of the epidemic. So Germany and China might offer reasonable choices but care needs to be taken as case-definitions varied considerably over time.

In relation to Australia, testing missed some cases initially but has been reasonably complete since mid-March. Unfortunately, no attempt is made to disaggregate domestic and imported cases here. As of the end of March, just under 80% of identified cases were considered to be imported (see [https://www1.health.gov.au/internet/main/publishing.nsf/Content/1D03BCB527F40C8BCA258503000302EB/\\$File/covid_19_australia_epidemiology_report_9_reporting_week_ending_23_59_aedt_29_march_2020.pdf](https://www1.health.gov.au/internet/main/publishing.nsf/Content/1D03BCB527F40C8BCA258503000302EB/$File/covid_19_australia_epidemiology_report_9_reporting_week_ending_23_59_aedt_29_march_2020.pdf))

It's unclear whether the model is producing similar results to this but I suspect the ratio of local to imported cases is somewhat different.

I think these issues need to be addressed in some way in order for simulations to be considered more realistic. The other issues that concern me are with validation where it's suggested that the 90% distance scenario fits the post-March data well ... this is not necessarily clear as for instance this clearly does not match the incidence series well (which is likely to be the best measured of the relevant statistics).

This brings me to (iii). For intervention guidance from models to be taken seriously in a policy context, the model has to be believable, to reproduce known behaviour and to be easily linked with measurable outcomes in terms of the policy levers. In regards to the first 2 points - the model seems believable but calibration needs some work. In regard to the simulation of interventions - some of the key details for isolation and quarantine (timing, contact reduction) needs more explanation and justification (and perhaps revision). School closure seem more convincing, although again some of assumptions around parents staying home and contact modification could do with better justification.

How one links the social distancing levels used here with data and the post-intervention case series seems a little more problematic. There are now a significant number of mobility data time series available - most of which produce much lower changes in mobility than citymapper. It also seems to be the case that interventions in Australia have had rather more effect on cases than the model here suggests, despite what appear to be lower levels of reduction in travel to workplaces etc.

So to my mind, there is need for some expansion of how social distancing is looked at in the model - perhaps in particular, how non-home contacts are modified and the dual role of modification in this variable as well as non-attendance at work.

Otherwise, readers are left with a fairly precise definition of a concept (social distancing) which produces highly differentiated effects over a smallish parameter range but where the occurrence of this intervention in practice may not closely match the simulations.

An additional significant issue with the article is that its structure and clarity leave quite a bit to be desired. At present the paper features elements of both narrative and structured papers and I think they should really choose one or the other.

As examples:

- (i) The importance of a higher asymptomatic fraction in children is emphasised in the abstract and discussion yet merits about 10 words in results.
- (ii) Much of the 1st paragraph of results belongs in the introduction and the 2nd paragraph belongs in methods.
- (iii) The "Intervention strategies" section really would greatly benefit from a table that summarises the interventions and key parameters (including for example timing for isolation and quarantine). The section is frankly overlong and could be easily condensed with greater clarity.
- (iv) It's somewhat unclear to me whether methods is intended to be in the main text or entirely within the supporting information. If the latter, some critical aspects of methods need to be brought

into the main text to allow sufficient understanding of how the interventions work.

(v) The figures should really have date x-axes - given that the paper is supposed to approximately match the Australian epidemic, it's not ok to just have a "number of days" axis represented.

(vi) Figure 1 does not define the legend elements in the caption. In addition cumulative incidence graphs should be matched to attack rates as %s rather than numbers (given that %s are at times quoted in the text). This applies to Figs 5 and 6 as well.

(vii) Units need to be stated consistently - e.g. growth rates are daily but this is often omitted.

(viii) Why does figure 3 show prevalence rather than say incidence which is a more measurable quantity (and the focus of more of the model comparisons).

(ix) In Figure 4, why does the $SD=0.7, SC$ line start to trend up prior to the end of the intervention window? Could this imply that the SC period was actually only 49 days in this simulation?

Also, the window period is apparently shifted about 5 days left of Fig 3 - why is this ... is it just a graphing error?

(x) Please refer to the specific figure panel in text when describing graph features (i.e Fig 3a etc.).

(xi) There are a few issues with tense in the main text (ie. switching past to present etc.). There are also a few areas where the language needs to be tightened (e.g. talk about transport of "a virus" internationally - really talking about a human-human respiratory virus here), use "exposure" instead of "infection" and in relation to serial and generation intervals - serial interval is symptoms-symptoms, while generation interval is exposure-exposure.

There are a few minor issues with the SI section as well

(i) subscript error in line 5 under section C.

(ii) $p^{g_j \rightarrow i}$ is defined twice within about 3 lines

(iii) $f(n-n_j | j, i)$ needs to be defined.

(iv) Under D - the contact table occurs first not transmission.

We would like to thank the referees for reviewing our paper and providing constructive criticism and helpful comments. Please find attached a new version of the manuscript, revised according to the reviewers' suggestions, with the modifications marked in red (except for minor grammatical edits). Our specific point-by-point responses are below.

Reviewer #2

This paper is clearly topical, and clearly likely to have impact (indeed, I know that this work has already had impact in the Australian media and through advice to government). The authors adapt an existing epidemic model, developed previously by them, and apply it to the current coronavirus pandemic. The model incorporates detailed demographic information and is therefore tuned to the Australia context, but there are obviously implications for other similar nations. To adapt this model they need to carefully tune parameters and calibrate their model by ensuring that it provides appropriate age stratification. The model is thoroughly described and their work with the model is a carefully constructed experiment (which is the primary original contribution to Nature Communications) which allows the authors to infer the effectiveness of various control measures. I am in favour of publication.

Response: We thank the reviewer for this positive feedback.

- *statements in the introduction about 80-90% compliance imply a confidence interval. How? Similarly, the figure "almost four weeks on average" should be quantified with some statement of confidence.*

Response: We have computed confidence intervals for the key variables and outcomes, using bootstrapping with the bias-corrected percentile method. When the sample size is relatively small ($N = 20$), we also report the full empirical range of the variables.

- *one of the main conclusions of this paper is the effectiveness of "social" distancing. A simplistic way to model this would be to change the topology of the contact network (for example: <https://arxiv.org/abs/2004.10396> - for obvious reasons, I don't insist on citation, but I am interested in the authors opinion of the contrast). Some commentary on how the demographic method compares to network based approaches (which would imply the exponential growth rate becoming linear) would be interesting.*

Response: A brief comparison is added (last paragraph of Conclusions, p. 9).

- *this work is also now (as I write this) becoming doubly relevant in the Australian context as the government considers removing restrictions. Some discussion on the implications of this model for those scenario planning exercises would be useful. For example, which measures should be reversed and (potentially) in which order? Which have most "risk"?*

Response: This is now discussed (3rd paragraph, p. 9) in terms of two "exit" strategies: (i) a state by state elimination of local community transmissions (with the restrictions remaining for a longer duration, but achieving lower cases and greater public confidence), and (ii) controlled adaptation aimed at some minimal level of symptomatic cases within the health system capacity (with phased and adaptive lifting of restrictions).

- R_0 figures are given to 3 significant figures, and elsewhere data is reported up to 7 significant digits. This seems to be excessive, given the nature of the work and the assumptions driving the model.

Response: R_0 estimates, as well as other similar estimates, are now rounded to two (three, when necessary) decimal points throughout the study, e.g., Table 9.

- throughout, figures are illustrated with standard deviations, and from 10 simulations. However, I suspect that the distribution is skew and a (perhaps) 90% confidence interval might be more appropriate. Moreover, it is unclear to me that 10 simulations really is a reasonable expectation. I would've thought the computation would be achievable with much larger numbers of replicates.

Response: Yes, we agree, and having doubled the number of runs, we now trace 95% confidence intervals (CIs), constructed with bias-corrected bootstrapping, around epidemic curves. The number of runs ($N = 20$) is sufficient to produce relatively narrow CIs.

- In addition to the previous point, simulations throughout seem to be reported with very tight uncertainty estimates. If I understand it correctly, that uncertainty is due to random outcomes in the agent based simulation - and not parameter uncertainty. Given that the 90% confidence intervals of most of the key parameters is huge (R_0 in (1.15, 4.77) in one study), some further comment, or computation indicating the sensitivity of the simulation to these parameters (and hopefully its robustness) would be useful.

Response: Yes, this issue is now explicitly addressed by a proper sensitivity analysis aimed to estimate the parameter uncertainty (in addition to stochasticity of Monte Carlo simulation across multiple runs, which is captured by CIs). The sensitivity analysis is carried out by using the Morris method (the elementary effect method), described in a new subsection 6.3 (in Methods). The results are reported in two parts: (i) sensitivity of the model to variations in model parameters, summarised in Appendix D.1; and (ii) sensitivity of the model outcomes (social distancing) to changes in the micro-distancing parameters, summarised in Appendix D.2. The analysis shows robustness of the model and its findings.

- Fig. 2(a) seems to indicate an uptick in cases before the end of control - or immediately after. Given the 1-2 week latency period I find both these results rather surprising.

Response: We now clarify (caption to Fig. 2) that the start and end days of social distancing varies across stochastic runs, while the shaded area is fixed across all the runs and compliance levels. For 70% SD it so happens that the last day of suppression on average (141.4) occurred slightly earlier than the “fixed” duration (142 days), bringing about the observed uptick.

- The quote concerning the Imperial College result needs some further qualification - what is the expected impact on the total number of infections?

Response: We added a sentence (4th paragraph on p. 7) quantifying the extent required to build up collective immunity, under the assumptions adopted in the model.

- It would be nice if Fig. 3 could be extended with contours, and more than 10 simulations

Response: Yes, it is a nice suggestion, improving the clarity of the phase transition shown in the new Fig. 4 (now after 20 runs and 5% increments). We also slightly updated the corresponding description of phase transition on p. 8 (1st paragraph).

- *In the case of Australia (and a small number of other territories) the pertinent question is not what will we need to do to prevent/mitigate a "second wave", but rather what is the probability of elimination (of the virus). Comment.*

Response: This is a difficult question, and while we have discussed this issue (3rd paragraph, p. 9) in terms of two "exit" strategies, the probability of elimination does not appear to be high at this stage, practically anywhere in the world, for various social and political factors that lie outside of the paper's scope.

- *ABS is not defined on first usage (page 11)*

Response: Defined now (p. 4).

- *conclusions on school closure depend (I assume) on rather uncertain assumptions about the transmissibility and infectivity in young people. It would be good to better describe these limitations of the stated results.*

Response: We have mentioned these limitations explicitly (last paragraph, p. 8).

- *page 19: What Is X_i ? Do the authors mean X_i ? Similarly f equation (2) is not defined.*

Response: The typo is fixed, and function f explained.

- *I find the demarcation of Adult (>18) and Child (<19) to be somewhat irritating - what about someone who is 18.5!? Perhaps and \geq sign could be used!? I realise (I assume) that the model assigns age to be integer valued, but still....*

Response: We changed these inequalities as suggested (Tables 6 and 7). The age is assigned an integer value (this clarification is added to the captions).

- *What is Citymapper?*

Response: This is explained (2nd paragraph on p. 36), complemented by a comparison with other sources as well.

- *Table 4 doesn't make sense to me. Some of the of the numbers (percentages, not fractions) go up when I would expect them to go down. Perhaps I am missing something that could maybe be explained better - but please check the data.*

Response: The fractions summarised in this table (now Table 9) are now explained in more detail, in Appendix J (p. 37).

- *Similarly, given that SC has been a clear and controversial part of the Australian response, it is unexpected that it only features in one scenario.*

Response: The SC policy is considered with, and without, social distancing, and the different effects are discussed separately: section 3.3 (shown in Fig. 2), and in Appendix G (Figures 14–17).

Overall, the paper could do with more careful proof-reading.

Response: The text, including the abstract and supplementary materials, is modified throughout.

Reviewer #3

This paper uses a detailed agent-based model (ACEMOD) of COVID-19 transmission to investigate the use of control strategies against COVID-19 in the Australian context. The model is spatially explicit and uses various national datasets, including census and school-attendance data to establish social mixing layers including schools, households, workplaces etc. and has adapted from previous studies looking at pandemic influenza.

The authors look at the specific interventions of case isolation, contact quarantine, school closure and social distancing with various levels of compliance. They find a transition in epidemic outcomes with social distancing compliance from 70-90%, with 70% insufficient to stop epidemic growth and 90% providing the potential for local elimination.

The paper is of interest because it applies a model that facilitates individual-level assessment of interventions that affect mobility to a realistic context. If the conclusions from this study could be considered generalisable, this would be of high interest both in the Australian and international contexts.

Response: We thank the reviewer for this positive feedback.

In assessing this generalisability, I will focus on what I consider to be the critical aspects of applied epidemiological modelling in this context:

- (i) the degree to which assumptions regarding relevant parameters underpinning the model are reasonable*
- (ii) the degree to which the model reflects available epidemic data through the calibration process*
- (iii) the degree to which conclusions about interventions can inform policy.*

In relation to (i), the natural history parameters for the model seem reasonable. I suspect that transmissibility is in reality somewhat more peaked around the onset of symptoms (and declines more quickly afterwards) but I don't believe there are robust data as yet to confirm this.

In relation to asymptomatic cases - their relative infectivity is currently unknown (and really the definition of "asymptomatic" in the literature is a woolly concept at present). I think the assumed asymptomatic fraction is reasonable - there is probably more of a gradient of this with age but the step-function is an ok approximation here. The generation interval for simulations here (~6.4 days) is similar to the earlier data from China and the assumptions used in the imperial college reports - there is considerable debate about this value, with a number of studies finding values in the range 4-5 days. However, these have tended to reflect settings in which mitigation is occurring which can be expected to contract the generation interval. So I feel this is reasonable. The values of R_0 used here are in the range 1.6 to 2.8, which cover a roughly appropriate range.

Response: We agree with this assessment, and added more recent references, confirming the adopted parameter ranges. We have also added an outlier removal technique (last paragraph on p. 24 and 1st paragraph on p. 25) to our stand-alone method of estimating R_0 and the generation time T_{gen} using multiple simulation runs. This resulted in changes of the mean values of two output variables: $R_0 = 2.77$ and $T_{gen} = 7.62$, both aligned with observations reported in recent literature (see p. 15, 1st paragraph). This adjustment does not change the underlying model which is driven by the input parameters.

The other element requiring parameterisation are the interventions. Part of the trouble here is that the main text does not necessarily highlight the relevant - e.g. the description of seeding is buried at the bottom of appendix section E and indicates the 0.04% of arrivals generate a new infection. In truth this was non-constant and growing from a much lower value initial to likely significantly above this by the time travel restrictions were initiated - how does this compare with the actual data on importations?

Response: We brought the specific consideration of "seeding" forward, and discussed the local community transmissions (which should be contrasted with both imported and unknown-origin cases),

using a comparison with the genomic surveillance results reported elsewhere. This is now described in Section 6.2 (p. 16). We discuss this issue later on, in response to a related comment below.

We also provided Table 1 (p. 5), delineating macro- and micro-distancing parameters (thank you for pointing this distinction). The description in Section 2 is now better structured.

In relation to isolation and quarantine - I could not find a clear description of when isolation occurred. Is this the day that symptoms occur? This would be unrealistic on average particularly for mild cases. I also think that basing the assumptions on the Imperial model is mistaken - in the context of the UK, wide-spread testing was not available. I think while those assumptions might be reasonable if your diagnosis is unknown, both cases and contacts would be expected to more strongly alter their behaviour in response to a known diagnosis.

Response: We provided a justification for the assumption that even relatively mild symptomatic cases are identified and isolated on pp. 3-4 (1st paragraph on p. 4). The assumptions made in our model are not based on those adopted by the Imperial College model, and we removed this line of argumentation.

The assumptions about parents staying home also seem unrealistic for school closure. While there may be no set legal age at which children must be supervised at home, it would not be considered reasonable to leave primary school age-children home alone, particularly in circumstance where they are supposed to have supervised learning. I may have mis-interpreted the parameterisation here but it really needs to be spelled out clearly and to be more clearly justified in terms of choice.

Response: We provided more details (subsection 2.5 on p. 4), explaining that the adopted parameter range is concordant with the report of The Australian Bureau of Statistics (ABS). The upper considered limit is a half of parents, 50%, staying home during school closures – this ensures adequate parental supervision.

In relation to changes in contact behaviour - are these percentages calculated on a daily basis (rather than a 12-hour cycle basis?). Would be good to clarify that.

Response: This is now clarified (12-hour cycle), in an expanded explanation of simulation cycles (subsection 2.6 on p. 4).

In regard to social distancing - it concerns me that this is handled via a single parameter. Surely here was an opportunity to look at how compliance, together with the reduction of non-household contacts (and potentially even household contacts) contributed? Australian guidance related to both micro and macro distancing with visitation of other households etc. excluded for a significant period.

Response: There is a number of parameters describing social distancing, and as mentioned above, we now summarise all macro- and micro-distancing model parameters in Table 1 (p. 5). We also carried out a sensitivity analysis of the model outcomes with respect to changes in the micro-distancing parameters, presented in section in Appendix D.1.

Also, in regard to combination strategies - are there interactions between the contact modifications that need to be considered when combining SC and social distancing. It seems that there is potential for rates of transmission to rise under school closure combined with social distancing and while possible, I'm not sure that actually reflects reality.

Response: There are some restrictions on possible interactions across contact modifications, and the dependencies are described in subsection 2.6 on p. 4. In general, these modifications are applied in a specific order, without “stacking”.

In relation to (ii) calibration - this is always a challenge for agent-based models. I have three main comments here

(i) Calibration is a critical part of a reader's acceptance of the model. The fact that there are no calibration results presented in the main text is a big issue for me. Figures 5 and 6 do not present calibration results ... they present intermediate results that can roughly be compared against data that is not shown in the figures. Ideally at the start of the results there would be a comprehensive comparison of the model to the data used in informing it.

Response: We now describe the calibration process in a more structured way, specifying five input parameters and four output variables – this is presented in Section 6.1. The estimates of the output variables and their confidence intervals are summarised in the 2nd paragraph on p. 16, with all variables attaining values close to our calibration targets. In addition, in Table 2 (p. 16) we detail the investigation of the ranges of R0 and generation time T_gen, driven by a separate model parameter (the scaling factor kappa).

(ii) I am unconvinced by the data used to roughly calibrate growth rates. Firstly, cumulative incidence is a poor measure - this is inherently biased by the initial observations, which are almost always inaccurate because surveillance plays catch-up with the epidemic. This concern can be applied to Italy, the UK and the US and to a lesser extent Germany. South Korea is not a good example because of the cluster nature of the initial growth and Iran's surveillance has many fundamental problems that mean it is not a good example for measurement of the epidemic. So Germany and China might offer reasonable choices but care needs to be taken as case-definitions varied considerably over time.

Response: We added an explanation, on p. 15 (3rd paragraph), for our choice of the daily growth rate of cumulative incidence as a calibration target. While in general we agree with the reviewer's points, we point out that, at the time, this variable was more accessible than other candidates, as we argue in the added text. It also provided a consistent calibration target across a number of countries, in addition to Australia, supporting our choice. We do agree that South Korea is not a relevant example, and it was excluded from the list when we identified the target range.

Importantly, we point out (last paragraph of subsection 6.1) that, in hindsight, other calibration targets can be chosen, refining the model. However, the model presented in this study was calibrated by 24 March 2020, and was used in accurately predicting the epidemic peaks in Australia in early April (both incidence and prevalence), while providing timely advice on relevant pandemic interventions. Changing the model now, in a “post-diction” mode, does not appear to us as a viable approach, and so we stay with the original parametrization.

In relation to Australia, testing missed some cases initially but has been reasonably complete since mid-March. Unfortunately, no attempt is made to disaggregate domestic and imported cases here. As of the end of March, just under 80% of identified cases were considered to be imported (see [https://www1.health.gov.au/internet/main/publishing.nsf/Content/1D03BCB527F40C8BCA258503000302EB/\\$File/covid_19_australia_epidemiology_report_9_reporting_week_ending_23_59_aedt_29_march_2020.pdf](https://www1.health.gov.au/internet/main/publishing.nsf/Content/1D03BCB527F40C8BCA258503000302EB/$File/covid_19_australia_epidemiology_report_9_reporting_week_ending_23_59_aedt_29_march_2020.pdf)) It's unclear whether the model is producing similar results to this but I suspect the ratio of local to imported cases is somewhat different.

Response: Section 6.2 (p. 16) now addresses the issue of local transmission directly, by comparing fractions of the local transmission detected by our model with the genomic surveillance results reported by Rockett et al. [new reference 25]. The comparison shows a good agreement between the model and genomic analysis. We point out that a significant proportion of cases fall neither in the category of locally acquired cases (25.8% identified by the genomic analysis, and 34.9% detected by our model), nor in the category of imported cases, being of unknown origin or under investigation.

The report cited above by the reviewer states that 79% of the “cases with a reported place of acquisition” had a recent international travel history – however, this computes into only 56.6% of all cases (as some sources of acquisition are unknown), and includes a significant proportion of passengers arriving on cruise ships. The study of Rockett et al. [25] observes that, in New South Wales, “Between 1 and 21 March 2020, the weekly proportion of imported cases was between 5 and 20%. During the same period, cases epidemiologically defined as ‘unknown origin/under investigation’ increased from none during the week beginning 1 March 2020 to between 31 and 35% from 8 to 21 March 2020”.

Thus, the fractions of local transmissions captured by our model are in concordance with the genomic analysis (at least for New South Wales, which represented 44.5% of all cases detected nationally, by 28 March 2020). Hence, we believe that the flow of overseas passengers arriving via air traffic, explicitly modelled by our study, is a substantial component for the pandemic “seeding”. We do accept that, in hindsight, we could have added maritime traffic. This is now acknowledged in Appendix H.1 (1st paragraph of the subsection, p. 34), in explaining the discrepancy in the size of incidence peak: for example, 10% of total cases on 18 April 2020 have been linked to the Ruby Princess cruise ship from which 2,700 passengers were allowed to disembark on 19 March.

Overall, the attribution of cases to different categories continues to be investigated in Australia, and we limited our discussion of the topic to subsections 6.2 and H.1.

I think these issues need to be addressed in some way in order for simulations to be considered more realistic. The other issues that concern me are with validation where it's suggested that the 90% distance scenario fits the post-March data well ... this is not necessarily clear as for instance this clearly does not match the incidence series well (which is likely to be the best measured of the relevant statistics).

Response: In a new subsection describing our forecasting results, H.2 (pp. 34-35), we provide more sources supporting that the compliance level of 90% has been achieved in Australia. These sources extend beyond Citymapper service, and include surveys by Vodafone and Australian Bureau of Statistics, as well as a national online survey reported elsewhere [new reference 72].

Having argued this point, we do acknowledge (1st paragraph of H.1, p. 34) that the actual levels of distancing vary across time, and in reality have been complemented by other surveillance, interventions, e.g., hotel quarantine of international arrivals, inter-state border closures, etc., which are not part of our model. Thus, the discrepancies between the model predictions and actual epidemic curves are expected. Nevertheless, we find the model’s performance over a three-month horizon quite satisfactory, with accurate predictions of the timings of both incidence and prevalence, and close predictions of the cumulative incidence at the end of suppression period.

This bring me to (iii). For intervention guidance from models to be taken seriously in a policy context, the model has to be believable, to reproduce known behaviour and to be easily linked with measurable outcomes in terms of the policy levers. In regards to the first 2 points - the model seems believable but calibration needs some work. In regard to the simulation of interventions - some of the key details for isolation and quarantine (timing, contact reduction) needs more explanation and justification (and perhaps revision). School closure seem more convincing, although again some of assumptions around parents staying home and contact modification could do with better justification.

Response: We thank the reviewer for this systematic analysis, and provided more details on these aspects, including sensitivity analysis, calibration, and details of the interventions, as described above.

How one links the social distancing levels used here with data and the post-intervention case series seems a

little more problematic. There are now a significant number of mobility data time series available - most of which produce much lower changes in mobility than citymapper. It also seems to be the case that interventions in Australia have had rather more effect on cases than the model here suggests, despite what appear to be lower levels of reduction in travel to workplaces etc.

So to my mind, there is need for some expansion of how social distancing is looked at in the model - perhaps in particular, how non-home contacts are modified and the dual role of modification in this variable as well as non-attendance at work.

Otherwise, readers are left with a fairly precise definition of a concept (social distancing) which produces highly differentiated effects over a smallish parameter range but where the occurrence of this intervention in practice may not closely match the simulations.

Response: In addition to the mobility data sources that we now consider (mentioned in the response to the previous point), we strengthen the study of social distancing by sensitivity analysis. Specifically, the second part of our sensitivity analysis explicitly addressed the question of how social distancing outcomes depend on the underlying micro-distancing parameters (Appendix D.2). This analysis shows that both the 90% SD scenario and the transition between 70% and 80% SD are robust to changes in the non-home contacts, as well as the household contacts. In addition, this analysis identifies a range of model applicability in terms of these parameters.

An additional significant issue with the article is that its structure and clarity leave quite a bit to be desired. At present the paper features elements of both narrative and structured papers and I think they should really choose one or the other.

Response: The structure of the paper has been significantly revised, with seven new tables, and more structured material presentation. We shortened some narrative, making it more structured – in the main text, and Methods, as well as in Supplementary Materials. In particular, some paragraphs were moved from Introduction to Appendix A, while sections 2 and 6 were substantially restructured.

As examples:

(i) The importance of a higher asymptomatic fraction in children is emphasised in the abstract and discussion yet merits about 10 words in results.

Response: This is now described in more detail in Section 6.1.

(ii) Much of the 1st paragraph of results belongs in the introduction and the 2nd paragraph belongs in methods.

Response: Re-arranged as suggested.

(iii) The "Intervention strategies" section really would greatly benefit from a table that summarises the interventions and key parameters (including for example timing for isolation and quarantine). The section is frankly overlong and could be easily condensed with greater clarity.

Response: Table 1 is added, and some paragraphs and sentences are removed.

(iv) It's somewhat unclear to me whether methods is intended to be in the main text or entirely within the supporting information. If the latter, some critical aspects of methods need to be brought into the main text to allow sufficient understanding of how the interventions work.

Response: Methods section is meant to stay together with the main text, while all the appendices are planned for Supplementary Materials.

(v) *The figures should really have date x-axes - given that the paper is supposed to approximately match the Australian epidemic, it's not ok to just have a "number of days" axis represented.*

Response: Date x-axis was added for relevant figures, with a note that the alignment between the days and the dates varies across the runs, being synchronised by crossing the threshold of 2,000 cases in each run.

(vi) *Figure 1 does not define the legend elements in the caption. In addition cumulative incidence graphs should be matched to attack rates as %s rather than numbers (given that %s are at times quoted in the text). This applies to Figs 5 and 6 as well.*

Response: We modified the captions to achieve consistency. Cumulative incidence is reported in “case numbers” throughout the text.

(vii) *Units need to be stated consistently - e.g. growth rates are daily but this is often omitted.*

Response: The growth rate is now reported as a fraction per day, and other adjustments have been made in legends and labels in various figures.

(viii) *Why does figure 3 show prevalence rather than say incidence which is a more measurable quantity (and the focus of more of the model comparisons).*

Response: “Active” cases (prevalence) continue to be a focus point for both modelling and policy setting with respect to the ongoing demand for, and strain on, healthcare. On the one hand, prevalence is chosen here to capture the more lasting effects than those captured in (often noisy) incidence data. On the other hand, prevalence captures information that is more relevant for adaptive policy setting than the information contained in the (often outdated) cumulative incidence. However, the phase transition can be observed with either alternative.

(ix) *In Figure 4, why does the $SD=0.7, SC$ line start to trend up prior to the end of the intervention window? Could this imply that the SC period was actually only 49 days in this simulation?*

Response: The SC period coincides with the SD period, and so this feature (trending up of the $SC + SD = 0.7$) is explained by the transient nature of the SC effect on compensating the lack of SD. This effect wears off when the transmissions driven by school-age children staying at home but socially mixing within the community and households accumulate to the levels comparable to their normal interactions in schools. The transient nature of this effect is now explicitly stated in Appendix G (1st paragraph on p. 34), and in the caption of new Fig. 17.

Also, the window period is apparently shifted about 5 days left of Fig 3 - why is this ... is it just a graphing error?

Response: Yes, it was a graphing error, fixed now.

(x) *Please refer to the specific figure panel in text when describing graph features (i.e Fig 3a etc.).*

Response: Fixed.

(xi) *There are a few issues with tense in the main text (ie. switching past to present etc.). There are also a few areas where the language needs to be tightened (e.g. talk about transport of "a virus" internationally - really talking about a human-human respiratory virus here), use "exposure" instead of "infection" and in relation to*

serial and generation intervals - serial interval is symptoms-symptoms, while generation interval is exposure-exposure.

Response: This has been hopefully addressed.

There are a few minor issues with the SI section as well

(i) subscript error in line 5 under section C.

(ii) $p^{g_j \rightarrow i}$ is defined twice within about 3 lines

(iii) $f(n_j | j, i)$ needs to be defined.

(iv) Under D - the contact table occurs first not transmission.

Response: These mistakes are now fixed.

Thanks again for your consideration.

REVIEWERS' COMMENTS

Reviewer #2 (Remarks to the Author):

The others have done a very good job in responding to my comments. I have no further corrections or suggestions and recommend publication in Nature Communications.

My only suggestion is self-serving: ref. 40 has now been published as IEEE Access, vol. 8, pp. 109719-109731, 2020, doi: 10.1109/ACCESS.2020.3001298.

Reviewer #4 (Remarks to the Author):

I did not review the initial draft of this paper. I am generally happy with the paper and it appears to be well thought out. I am restricting my attention to things that are relatively quick changes in light of the timescale.

I am a little surprised by how tight the confidence intervals are reported in the paragraph before 3.1. Can the authors comment on this? This makes me a bit nervous.

Also, I don't see a discussion of how results would change if the data input into the model had many more infections in children. I think there is a clear danger that many children are undiagnosed. Remarkably, I still haven't seen this question resolved in the data that people have collected. So I am concerned that attempts to fit to measured data may force the model to underestimate the potential role of children. How would this change outcomes, particularly about school closure?

We would like to thank the editor and referees for reviewing our paper and providing helpful comments. Please find attached a new version of the manuscript, revised according to the suggestions, with the modifications marked in **red** (changes) and **blue** (moved text). Our specific point-by-point responses are below.

Reviewer #2

My only suggestion is self-serving: ref. 40 has now been published as IEEE Access, vol. 8, pp. 109719-109731, 2020, doi: 10.1109/ACCESS.2020.3001298.

Response: Thank you, we updated the reference.

Reviewer #4

I am a little surprised by how tight the confidence intervals are reported in the paragraph before 3.1. Can the authors comment on this? This makes me a bit nervous.

Response: We now provided an explanation (last paragraph on page 3) differentiating between the confidence intervals reflecting the stochastic simulation and a broader range of possible variations in response to changes in the input parameters. The latter aspect is studied with both local and global sensitivity analysis (Morris method), presented in Appendix D (Supplementary Information).

Also, I don't see a discussion of how results would change if the data input into the model had many more infections in children. I think there is a clear danger that many children are undiagnosed. Remarkably, I still haven't seen this question resolved in the data that people have collected. So I am concerned that attempts to fit to measured data may force the model to under-estimate the potential role of children. How would this change outcomes, particularly about school closure?

Response: We agree with this concern, and commented on it in Discussion (last sentence of 2nd paragraph, page 7): “As the clinical picture of COVID-19 in children continues to be refined [34], these findings may benefit from a re-evaluation when more extensive pediatric data become available.”, as well as in section 5.1 (Calibration, last paragraph, page 15), acknowledging “a limited testing capacity resulting in possible under-reporting of cases (especially pediatric)”.

Other marked changes were done to address editorial concerns, including structure and style. Thanks again for your consideration.